# Theory of Space: Can Foundation Models Construct Spatial Beliefs through Active Exploration?

**Pingyue Zhang**[1,*,†]**, Zihan Huang**[*]**, Yue Wang**[4,*]**, Jieyu Zhang**[3,*]**,**
**Letian Xue**[1]**, Zihan Wang**[1]**, Qineng Wang**[1]**, Keshigeyan Chandrasegaran**[2]**,**
**Ruohan Zhang**[2]**, Yejin Choi**[2]**, Ranjay Krishna**[3]**, Jiajun Wu**[2]**, Li Fei-Fei**[2]**, Manling Li**[1,†]

[1]Northwestern University   [2]Stanford University   [3]University of Washington   [4]Cornell University
pingyuezhang@u.northwestern.edu, manling.li@northwestern.edu

[*]Equal contribution   [†]Corresponding author

## Abstract

Spatial embodied intelligence under partial observability requires agents to actively acquire missing information rather than passively consume complete observations. While multimodal foundation models excel at passive perception and reasoning, their ability to support active, self-directed exploration to build and maintain a coherent spatial belief remains unstudied. We therefore propose THEORY OF SPACE, defined as an agent's ability to construct, revise, and exploit a spatial belief through self-directed active exploration under partial observability. We implement THEORY OF SPACE using a benchmark with textual and visual environments. Rather than solving specific tasks, the goal is curiosity-driven exploration to build a complete, accurate spatial belief. A core innovation is spatial belief probing: we prompt it to reveal its internal spatial belief as a cognitive map at each step, letting us measure the quality of its underlying spatial belief. Our evaluation of state-of-the-art models on a suite of downstream tasks reveals critical bottlenecks: (1) **The Active-Passive Gap**: Performance degrades when agents must autonomously gather information (e.g., GPT-5.2: 57.1→46.0); (2) **Inefficiency**: Models explore in an unsystematic way and with high redundancy, failing to match the efficiency of program-based proxies while producing no better results. Through belief probing, we diagnose that perception acts as an initial bottleneck, yet global beliefs suffer further from **instability** that causes spatial knowledge to degrade over time. Finally, a false belief paradigm reveals **Belief Inertia**: agents fail to overwrite obsolete priors, an effect especially severe in vision-based models[1].

## 1 Introduction

Spatial embodied intelligence relies on active exploration. Unlike disembodied systems that passively process fixed observations, an embodied agent could take actions to alter its position in the environment as *exploration*, selectively acquiring observations needed to construct spatial knowledge for various spatial tasks. Cognitive science shows that such active exploration leads to substantially better spatial understanding than passively receiving the same information, even when observations are identical (Held & Hein, 1963; Chrastil & Warren, 2012; 2013). But exploration isn't simply about collecting more observations. It is about efficiency, acting under uncertainty to target what is unknown or ambiguous in the agent's spatial belief and maximize information gain.

We propose THEORY OF SPACE as a framework that explicitly treats exploration as a first-class decision-making problem, decoupled from any single downstream task, focusing on opening the box of the agent's internal spatial belief. Just as Theory of Mind (ToM) measures how agents model the hidden mental states of others, THEORY OF SPACE assesses an agent's ability to model the hidden physical structure of the world. We define THEORY OF SPACE as an embodied agent's ability to actively **construct**, **revise** in a dynamic environment, and **exploit** an internal *spatial belief* formed through active exploration. Beyond end-task evaluation, THEORY OF SPACE directly probes what the agent knows, what remains uncertain, and how effectively its actions reduce those uncertainties,

---

[1]The code and data are publicly available at https://theory-of-space.github.io/

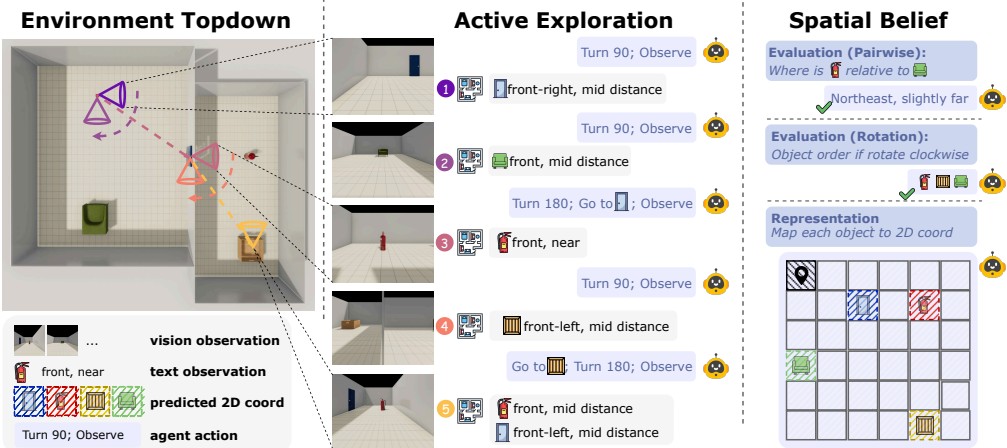

Figure 1: **THEORY OF SPACE: active exploration, belief probing, and evaluation.** Left: Agent trajectory under partial observability. Middle: Action loop across text/vision environments with egocentric observations. Right: Evaluation via spatial task exploitation and cognitive map probing.

measured by the number of exploration steps and the uncertainty resolved per action. Figure 1 provides an overview of THEORY OF SPACE's active exploration, belief probing, and end-task evaluation.

We apply THEORY OF SPACE to evaluate multimodal language models, which are promising candidates for embodied agents. By integrating vision and language, they support unified perception, reasoning, and action over time, yet existing foundation-model benchmarks offer little insight into these capabilities. Most current benchmarks fall into two categories: *passive* (Weston et al., 2015; Shi et al., 2022; Yang et al., 2025c; Gholami et al., 2025; Yang et al., 2025a), where the agent is only asked to reason over given observations, and *task-driven* (Gordon et al., 2018; Shridhar et al., 2020b; Li et al., 2025; Yang et al., 2025b), where the agent must achieve a specific goal (e.g., "*find the red chair*").

We propose to systematically evaluate the active process of spatial belief construction. Unlike passive benchmarks, our THEORY OF SPACE benchmark requires agents to actively explore via *moving*, *rotating*, and *observing* to build coherent global beliefs. We implement a scalable environment using ThreeDWorld (Gan et al., 2021) and Objaverse (Deitke et al., 2022) with *text-based* and *vision-based* worlds to localize perception versus reasoning failures. After exploration, we evaluate along two axes: **belief exploitation** via spatial downstream tasks probing *route*-level and *survey*-level knowledge (Siegel & White, 1975; Montello, 1998), and **exploration efficiency** via steps taken and information gain over time. We further design scripted proxy agents to disentangle exploration from reasoning. Our evaluation of state-of-the-art foundation models reveals promising capability in the text world but striking limitations in the vision world. **Active exploration remains a primary bottleneck**: models perform reasonably well passively but degrade when actively gathering information (e.g., GPT-5.2: 57.1 → 46.0; GEMINI-3 PRO: 60.5 → 57.3). Moreover, we also find a major efficiency gap: rule-based proxy agents reach target coverage in ∼9 steps, whereas foundation models explore redundantly, requiring ≥14 steps without improving belief accuracy.

Beyond downstream task scores, a core contribution of THEORY OF SPACE is **explicit cognitive-map probing**, which directly exposes the agent's evolving spatial belief during exploration rather than treating the model as a black box. This enables fine-grained diagnosis of *how* models represent space, revealing that while perception acts as an initial bottleneck, global beliefs further suffer from instability, causing knowledge to degrade over time. We also introduce a *False Belief* paradigm that alters the environment after initial exploration, uncovering spatial belief inertia: agents persistently retain obsolete spatial priors even after directly observing the updated configuration, revealing a critical failure in spatial memory revision.

An important direction for future work is to extend THEORY OF SPACE beyond single-agent settings to multi-agent exploration, where additional challenges arise around coordination and aligning (or sharing) spatial beliefs across agents.

## 2 THEORY OF SPACE

To build agents with spatial intelligence, we argue for evaluating not merely passive reasoning, but the **active, self-directed construction** of spatial belief from partial observations. We introduce THEORY OF SPACE, a conceptual counterpart to *Theory of Mind* (ToM). While ToM models hidden mental states of others, THEORY OF SPACE models **uncertain, currently unobserved** structure of space.

> 📄 **Definition: THEORY OF SPACE**
>
> Ability to **construct**, **revise**, and **exploit** an internal spatial belief.

Here, an **internal spatial belief** is a mental model (Taylor & Tversky, 1992) of spatial layout and relations maintained in working memory and updated from partial observations. We provide the formal definition of THEORY OF SPACE in Appendix A.1.

### 2.1 A PARADIGM FOR ASSESSING THEORY OF SPACE OF LARGE FOUNDATION MODELS

We propose a new paradigm for Assessing THEORY OF SPACE of large foundation models, which consists of three essential components below.

**Task-Agnostic Active Exploration.** Evaluating THEORY OF SPACE requires a shift from downstream task completion to exploration, where the agent decides *what to see next*. We place the agent in a partially observable environment and challenge it to actively select actions (*moving*, *rotating*, *observing*, and *terminating*) to build a general-purpose internal spatial model through self-directed exploration. This process encompasses both **Belief Construction** and **Belief Revision**. Inspired by the *false belief* paradigm in Theory of Mind (Wimmer & Perner, 1983) and *spatial belief revision* (Knauff et al., 2013), we evaluate whether agents can detect environmental changes and correctly revise their internal beliefs during exploration. The agent must identify what remains uncertain and terminate only upon acquiring sufficient evidence to form an accurate internal map.

**Belief Exploitation Assessment.** To translate THEORY OF SPACE into concrete evaluation tasks, we draw insights from the development of spatial representations (Siegel & White, 1975; Montello, 1998) and define two tasks to measure an agent's ability to exploit its internal belief for goal-directed behavior: (1) **Belief on Route** evaluates a path-based understanding of space organized around landmarks such as pairwise spatial relationships along a *egocentric* navigation; (2) **Belief on Survey** assesses a map-like "bird's-eye view" that represents space allocentrically, allowing for the inference of global relationships.

**Explicit Probing of the Internal Spatial Belief.** Behavioral success cannot directly reveal the quality of an agent's internal spatial model. We therefore require the agent to explicitly externalize its belief by *probing its cognitive map* at each turn. Cognitive maps are structured allocentric representations of space, well-established in neuroscience (Tolman, 1948; O'Keefe & Dostrovsky, 1971; Hafting et al., 2005), and serve as our canonical representation of hidden spatial structure. We evaluate map correctness and diagnose reasoning with dynamic signals capturing how reliably observations are integrated, tracked over time, and kept coherent across local and global structure. We further probe uncertainty modeling by requiring the agent to identify unobserved regions. This shifts evaluation from behavioral success to a direct assessment of representational competence.

## 3 BENCHMARKING THEORY OF SPACE ABILITY FOR FOUNDATION MODELS

Unlike task-driven benchmarks that only test task completion, we aim to answer "*can the agent form a global environmental belief through exploration?*". We structure the benchmarking into two phases. In the **Exploration Phase I**, the agent interacts with the environment to construct spatial belief by selecting and executing actions in the action space in § 3.1, and gather a sequence of local observations to integrate them into a unified spatial belief. In the **Reasoning Phase II**, the agent is asked to conduct spatial tasks (detailed in § 3.2).

### 3.1 SPATIAL ENVIRONMENT CONSTRUCTION

To ensure controlled experimentation, we procedurally generate multi-room indoor layouts on an $N \times M$ grid. Each scene is populated with $n$ indoor objects, each assigned a 2D integer coordinate

and a cardinal orientation from (`N`, `S`, `E`, `W`). The agent begins at a random position, is informed of the total number of rooms and the names of all objects in the scene, and then starts exploration. Following the Gym-style interface (Brockman et al., 2016), we define highly scalable environments in which each random seed deterministically instantiates a distinct multi-room layout.

**Action Space in the Environment.** The agent's interaction with the world is designed to focus on high-level decision-making rather than low-level motor control: `Goto` to move directly to a currently visible object; `Rotate` to turn in place by $90°$, $180°$, or $270°$; `Observe` to perceive visible objects in the $90°$ field of view; and `Query` to obtain a visible object's absolute 2D coordinates. We additionally assign a cost of 1 to `Observe` and 2 to `Query`, encouraging `Query` to be used only when necessary to resolve ambiguity. However, across all models `Query` is rarely invoked, so we only focus on `Observe` and measure exploration efficiency by step count instead of cost.

**Observation Feedback from a Text-Vision Parallel Environment.** We offer both text- and vision-based environments, enabling diagnostic analysis of spatial reasoning. Each `Observe` action returns both textual and visual feedback from a $90°$ field of view. The **Text World** provides symbolic observations with discrete bins for direction and distance (e.g., "chair is front-left and near", detailed below), isolating pure spatial reasoning. The **Visual World** instead supplies ego-centric RGB images rendered in ThreeDWorld (Gan et al., 2021) with Objaverse assets (Deitke et al., 2022), requiring perception to recover spatial relations. To calibrate perception in the visual setting, we provide two reference images, indicating unit distance (1 grid unit) / angle (a $22.5°$ angular cone), and showing all objects with their "front" orientation respectively. Details are shown in Appendix B.1

**Spatial Relation Representation.** To ensure that agents perceive and communicate about space using a consistent language across tasks and modalities, we discretize spatial relationships for directions and distances. For **allocentric direction**, we discretize into eight $45°$ bins aligned with the four cardinal and four intercardinal directions, denoted compactly as $\{$`N`, `NE`, `E`, `SE`, `S`, `SW`, `W`, `NW`$\}$. Each bin spans $45°$ around its heading (e.g., `N` $= [-22.5°, 22.5°)$). For **egocentric direction**, within a $90°$ forward field of view (FOV), we use five labels: `front-left` $[-45°, -22.5°)$, `front-slight-left` $[-22.5°, 0)$, `front` $0°$, `front-slight-right` $(0, 22.5°]$, and `front-right` $(22.5°, 45°]$. For **distance**, measured in map units independent of direction, we define six bins: *same* $= 0$, *near* $(0, 2]$, *mid* $(2, 4]$, *slightly far* $(4, 8]$, *far* $(8, 16]$, and *very far* $(16, 32]$.

## 3.2 DOWNSTREAM SPATIAL TASKS

We use open-ended questions rather than multiple-choice questions to reduce the risk of knowledge leakage. Drawing on prior work (Siegel & White, 1975; Montello, 1998), we define tasks to evaluate an agent's **Route** and **Survey** knowledge. Route belief captures how an agent encodes paths and spatial relations from an egocentric step-by-step perspective. Survey belief is a map-like, allocentric representation. Under **route belief**, the static task requires reporting direction and distance between two objects (*direction*); forward dynamics tasks ask the model to adopt an object's perspective (*persp.take*) or predict the final observation after executing a `Goto`/`Rotate` sequence (*act2view*); backward dynamics tasks infer which perspective is currently adopted (*perc.dec*) or recover the action sequence yielding a target observation (*view2act*). Under **survey belief**, the static task reconstructs global coordinates for all objects (*alloc.map*); forward dynamics tasks predict front-facing objects during a full self-rotation (*ment.rot*) or infer the observation from a given global pose (*loc2view*); the backward dynamics task localizes the agent from a target observation (*view2loc*). Details of these downstream tasks are shown in Appendix A.2.

## 3.3 ASSESSMENT DIMENSIONS

We define assessment dimensions that align with the core THEORY OF SPACE abilities: **construction** and **revision** are evaluated via exploration efficiency and belief quality, while **exploitation** is evaluated via task success. The detailed definition of assessment dimensions is shown in §A.3

**(D1) Belief Construction Efficiency.** *Measures how efficiently the agent collapses spatial uncertainty.* Quantified via normalized information gain (ratio of eliminated to total possible object positions, ranging from 0 to 1).

**Belief Representation and Quality Assessment.** We decompose the cognitive map into: **(D2) Cognitive Map (Observed)**, *measuring fidelity and coherent integration of observations*, evaluated

via Correctness (composite of positional, directional, and facing accuracy) and dynamic diagnostics; and **(D3) Uncertainty Map (Unobserved)**, *measuring how well the agent identifies unobserved regions*, evaluated via $F_1$ of *selecting* unexplored over candidate positions.

**(D4) Belief Revision.** *Measures the agent's ability to revise spatial beliefs under covert environmental changes.* Evaluated via the *False Belief* task (§5.2), scored by detection $F_1$ (identity and transformation type), with *Belief Inertia* quantifying bias toward obsolete priors.

**(D5) Belief Exploitation Success.** *Measures task success when the agent must utilize its spatial belief.* Relation-based tasks (direction, perspective-taking, action2view) score direction and distance separately (0.5 each); coordinate-output tasks (view2loc, alloc.map) use coordinate similarity.

### 3.4 EXPLORATION STRATEGIES

To rigorously evaluate spatial cognition, we distinguish between two capabilities: the ability to acquire information (exploration) and the ability to synthesize it (reasoning). We present two evaluation settings: (i) *Active Exploration*, where the agent must plan actions to reduce uncertainty, and (ii) *Passive Comprehension*, where the agent reasons over standardized logs generated by scripted proxies.

**Uncertainty-Driven On-Policy Exploration.** We conduct active evaluation to understand agent ability in **exploring the environment to gather necessary information in building spatial belief**. In this setting, the evaluated agent must plan and execute its own information-gathering policy. At each step, the agent selects an action based on its observation history and current objective, then receives new observations (text or image). Exploration continues until the agent issues an exploration *termination* or reaches the step budget. Success requires balancing two goals: maximizing coverage of unknown relations while minimizing action cost. This setting directly reveals whether the agent can recognize what it does not yet know and actively reduce uncertainty through exploration.

**Passive Exploration via Scripted Proxy Agents.** Evaluating THEORY OF SPACE also requires disentangling exploration quality from spatial reasoning ability. To isolate the latter, we introduce *proxy agents* that supply a fixed observation stream to evaluated models, eliminating variance from exploration failures and enabling fair comparison of core reasoning across architectures. We design two scripted proxies: SCOUT (visual environments) rotates exhaustively at each location to ensure full object coverage, and STRATEGIST (text environments) follows a belief-driven viewpoint selection policy to maximally reduce relational ambiguity. Implementation details appear in Appendix B.2.

## 4 EVALUATION AND ANALYSIS

We evaluate a set of state-of-the-art proprietary and open-source foundation models. They are evaluated on both passive and active settings described in § 3.4. Unless otherwise specified for ablations, all experiments use three connected $6 \times 6$ rooms with 4 objects in each (total 12 objects). To enable a like-for-like comparison between the text and vision settings, we instantiate identical room layouts across modalities. We use $384 \times 384$ images in the vision setting. We generate 100 scenes and create three questions per task per scene, yielding $3 \times 9 \times 100 = 2700$ questions per setting. We mainly evaluate six foundation models: GPT-5.2 (OpenAI, 2025), GEMINI-3 PRO (Google, 2025), CLAUDE-4.5 SONNET (Anthropic, 2025), GLM-4.6V (Zhipu AI Team, 2025), QWEN3-VL (Bai et al., 2025) (235B-A22B-Thinking), and INTERNVL-3.5 (Wang et al., 2025) (241B-A28B). For closed-source reasoning models GPT-5.2, GEMINI-3 PRO, and CLAUDE-4.5-SONNET, we set the temperature to 1 and the maximum number of tokens to 32768. For all other models, we set the temperature to 0. INTERNVL-3.5 supports at most 10 images, so we omit it for the vision-based world setting.

**Active Exploration Results.** We evaluate models as active agents, where they must autonomously explore the environment to build their spatial belief and terminate the exploration process by their own. This setting tests the full THEORY OF SPACE pipeline, requiring the agent to simultaneously plan an efficient information-gathering trajectory, integrate observations, and maintain a coherent cognitive map under uncertainty. The agent's performance is measured by its Exploration Efficiency as shown in § 3.3 and its final accuracy on the downstream spatial tasks. The agent has a maximum of 20 exploration steps. Table 1 presents the active performance of the models, providing a holistic view of their ability to translate curiosity into knowledge. Figure 2 illustrates information gain over the course of the exploration turns. GPT-5.2 acquires substantial information early on, but its rate

| Methods | Avg.step | direction | persp.take | perc.dec. | act2view | view2act | alloc.map | ment.rot | loc2view | view2loc | Avg. | Pass. Avg |
|---|---|---|---|---|---|---|---|---|---|---|---|---|
| | | Static (S) | Dynamic (D) | | | | Static (S) | Dynamic (D) | | | | |
| | | Route | | | | | Survey | | | | | |
| **Vision-based World** | | | | | | | | | | | | |
| *Proprietary Models* | | | | | | | | | | | | |
| GPT-5.2 | 17.2 | 40.0 | **36.7** | 56.2 | 43.8 | 40.3 | 43.4 | 59.7 | 56.9 | 37.8 | 46.0 | 57.1 |
| GEMINI-3 PRO | **13.6** | **56.3** | **36.7** | **68.2** | **47.2** | **54.0** | **63.5** | **73.0** | **65.4** | **52.2** | **57.3** | **60.5** |
| CLAUDE-4.5 SONNET | 19.6 | 23.7 | 23.3 | 18.7 | 33.3 | 10.7 | 37.4 | 34.7 | 33.7 | 50.9 | 29.6 | 43.1 |
| *Open-source Models* | | | | | | | | | | | | |
| GLM-4.6V | **15.0** | 15.8 | 18.5 | 3.3 | 14.0 | 0.7 | 18.9 | 8.0 | 18.5 | 31.8 | 14.4 | 16.7 |
| QWEN3-VL | 16.3 | **16.8** | **23.3** | **13.4** | **24.8** | **5.7** | **25.8** | **16.3** | **21.5** | **43.7** | **21.3** | **24.9** |
| HUMAN | 9.8 | 94.5 | 100.0 | 100.0 | 100.0 | 93.4 | 93.4 | 100.0 | 100.0 | 86.7 | 96.4 | - |
| HUMAN WITH TOOL* | 11.1 | 100.0 | 100.0 | 100.0 | 100.0 | 97.8 | 100.0 | 100.0 | 100.0 | 93.4 | 99.0 | - |
| **Text-based World** | | | | | | | | | | | | |
| *Proprietary Models* | | | | | | | | | | | | |
| GPT-5.2 | **11.4** | 68.8 | 70.5 | 80.3 | 71.0 | 53.7 | 77.9 | 81.0 | 79.1 | 66.0 | 72.0 | **90.4** |
| GEMINI-3 PRO | 13.5 | **78.0** | **79.2** | **90.6** | **75.3** | **76.3** | **81.0** | **94.0** | **83.3** | **76.2** | **81.5** | 86.5 |
| CLAUDE-4.5 SONNET | 18.7 | 65.3 | 65.3 | 79.0 | 62.7 | 51.7 | 68.8 | 76.3 | 57.0 | 67.0 | 65.9 | 73.6 |
| *Open-source Models* | | | | | | | | | | | | |
| GLM-4.6V | 14.5 | 20.8 | 19.7 | 12.7 | 21.8 | 3.7 | 13.9 | 9.3 | 22.7 | 26.2 | 16.8 | 23.4 |
| INTERNVL-3.5 | 15.0 | 28.8 | 44.8 | 26.0 | **36.8** | 7.3 | 31.0 | 27.7 | 33.8 | 38.9 | 30.6 | 37.4 |
| QWEN3-VL | **14.1** | **32.3** | **45.7** | **48.2** | 33.3 | **11.7** | **36.4** | **34.7** | **35.7** | **49.9** | **36.8** | **45.6** |
| HUMAN | 10.8 | 87.8 | 82.1 | 100.0 | 85.5 | 86.8 | 66.6 | 100.0 | 95.6 | 75.8 | 86.7 | - |
| HUMAN WITH TOOL* | 12.8 | 100.0 | 100.0 | 100.0 | 100.0 | 100.0 | 100.0 | 100.0 | 100.0 | 91.2 | 99.0 | - |

Table 1: **Exploitation Performance** (%) **of Belief Construction via Active Exploration.** Models autonomously plan actions and are evaluated on exploration cost and reasoning across text- and vision-based environments. GEMINI-3 PRO leads almost every task. Humans outperform in both settings, especially in vision. *Humans can use instruments such as protractors and compasses to infer object positions precisely. We also include passive (*pass.*) results here for comparison.

of gain slows in later turns, resulting in lower cumulative information gain than GEMINI-3 PRO and CLAUDE-4.5 SONNET. Moreover, none of the models achieves full coverage relative to the proxy agent. We benchmarked three human subjects across five text and five vision scenes. Humans consistently outperformed foundation models in both domains, particularly in vision. Intuitively, humans scored higher in vision than text as visual information is easier to process. With tools, they achieved near-perfect accuracy.

**Passive Exploration Results.** We evaluate models on trajectories generated by rule-based proxy agent to understand a model's core spatial reasoning ability regardless of its exploration strategy. The last column of Table 1 reports the average performance of each model in both text-based and vision-based environments. The detailed experimental results are shown in Appendix C.1. As evaluated, the results show a clear separation: GPT-5.2 and GEMINI-3 PRO lead by a wide margin over other systems, particularly open-source models. A substantial **modality gap** persists, with text performance far better than vision performance for all models.

> 💡 **Key Findings: Modality Gap**
> - **Modality Gap Exists:** text significantly outperforms vision.

| | Text-based World | | | | | |
|---|---|---|---|---|---|---|
| Methods | 2-room | | | 4-room | | |
| | *pass.* | *act.* | *steps* | *pass.* | *act.* | *steps* |
| GPT-5.2 | **92.3** | 77.8 | **6.2** | **86.5** | 66.0 | **16.4** |
| GEMINI-3 PRO | 86.7 | **80.6** | **6.2** | 81.2 | **77.7** | 19.7 |
| | Vision-based World | | | | | |
| Methods | 2-room | | | 4-room | | |
| | *pass.* | *act.* | *steps* | *pass.* | *act.* | *steps* |
| GPT-5.2 | **59.3** | 51.5 | 10.8 | 52.6 | 40.3 | 23.2 |
| GEMINI-3 PRO | 58.3 | **57.8** | **6.6** | **56.2** | **51.5** | **19.7** |

Table 2: **Exploitation Performance** (%) **for Multi-Room Settings** (2-room and 4-room). *pass.* for passive avg acc, *act.* for active avg acc, *steps* for average steps.

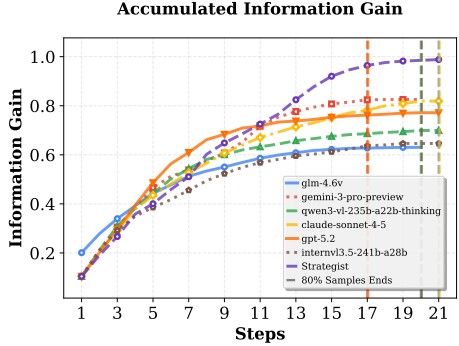

Figure 2: **Accumulated information gain** over exploration steps in the text world.

Overall, active accuracies underperform the passive setting. As shown in Figure 2, models such as GPT-5.2 gather information quickly but terminate prematurely, leaving unresolved uncertainty. Compared to the STRATEGIST proxy, which achieves full certainty, autonomous models remain less thorough. A second disparity is efficiency. In the vision domain, the SCOUT proxy reaches target coverage in $\approx 9$ steps, whereas autonomous models expend significantly more actions with no performance benefit. In the text domain, following the concise SCOUT trajectories ($\approx 9$ steps), GPT-5.2 and GEMINI-3 PRO achieve accuracies of 83.9 and 86.7, surpassing their active exploration scores (72.0 and 81.5, Table 1). This confirms that models perform better when guided by efficient proxy paths than when exploring autonomously.

**Different Room Settings.** For the two best-performing models, GPT-5.2 and GEMINI-3 PRO, we further evaluate performance under multi-room configurations (one four-room and two three-room settings). As shown in Table 2, performance declines and the active–passive gap widens as room count increases. Notably, GEMINI-3 PRO requires nearly the same exploration steps in text-only and vision-based environments. Detailed results are in Appendix C.2.

> 💡 **Key Findings: Active Exploration as the Bottleneck**
>
> - **Performance and Efficiency Deficit:** Active agents score lower than reasoning on rule based program histories, and explore less efficiently than the program.
> - **Incomplete Coverage:** Active agent fails to achieve complete information coverage.
> - **Complexity-Widened Gap:** The active versus passive difference grows with environment scale; GEMINI-3 PRO degrades least.

**Exploration Pattern.** Manual inspection reveals distinct behavioral patterns. GPT-5.2 explores unsystematically, prioritizing newly discovered doors and leaving rooms partially unexplored, resulting in object omission and path redundancy. GEMINI-3 PRO adopts a methodical "rotate-and-scan" strategy before transitioning between rooms, mirroring the SCOUT proxy agent. Further examples are provided in Appendix D.2.

# 5 HOW DO FOUNDATION MODELS MANAGE INTERNAL SPATIAL BELIEF?

In this section, we use the THEORY OF SPACE belief-probing mechanism (as proposed in §2.1) to diagnose how MLLMs manage internal spatial beliefs and move beyond treating the agent as a black box. Figure 3 shows the example of how we probe the belief of agent at each exploration step.

## 5.1 COGNITIVE MAP PROBING

Instead of treating the spatial belief as a black box, we probe the agent's internal state to distinguish verifying known facts from hypothesizing about the unknown. The agent externalizes its belief via a structured JSON containing a **Cognitive Map**, which records observed objects.

**Representation.** For cognitive map, the agent presents its belief as a single, allocentric cognitive map serialized in structured JSON. The map maintains (i) a *global* layout anchored to the agent's initial pose, and (ii) a *local* snapshot that records only the currently visible objects with the current pose as origin to diagnose immediate perceptual errors.

**Metrics.** We evaluate cognitive maps along three axes: *positional accuracy* (Euclidean similarity between predicted and true coordinates, scaled by object coverage), *directional accuracy* (pairwise directional relation accuracy), and *facing accuracy* (fraction of correctly predicted object orientations). Using global and local belief representations updated at each turn, we compute five diagnostic scores: **Correctness** (composite accuracy of the final-turn global map); **Perception** (accuracy of local map interpretation for newly observed objects in the current FOV); **Self-tracking** (agent pose estimated from the predicted global map vs. ground truth); **Local↔Global consistency** (agreement between local and global predictions within the same turn); and **Stability** (whether beliefs about previously observed objects remain non-degrading across turns). Full metric definitions are provided in Appendix A.4.

Table 3 reveals a substantial modality gap: performance drops markedly in the vision setting across all metrics. **Self-tracking** is not a primary bottleneck, as models generally maintain accurate pose

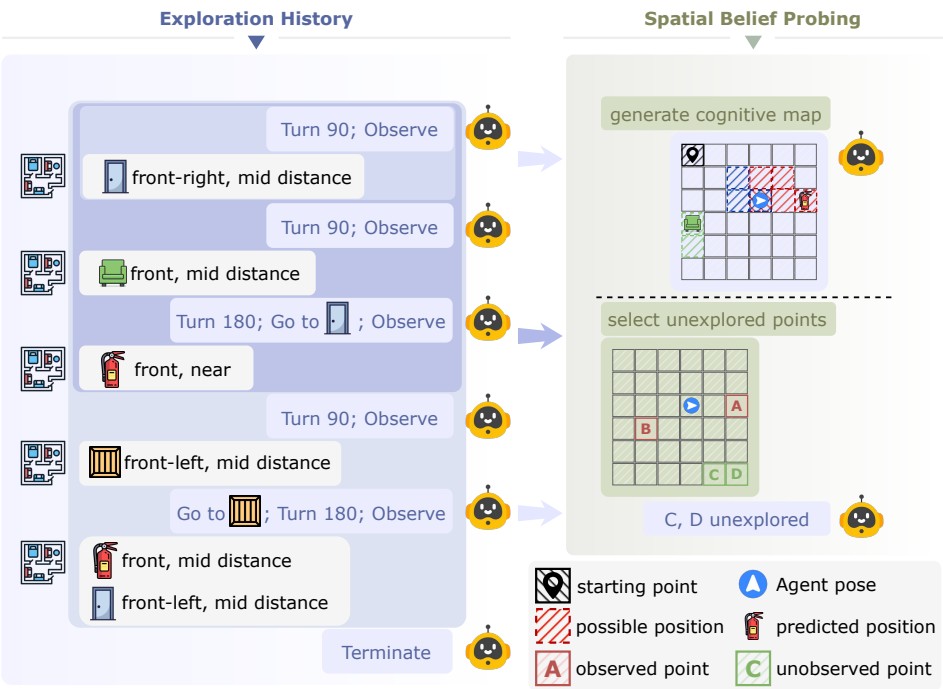

Figure 3: **Internal Spatial Belief Probing.** At each step, the agent executes an action, receives an observation, and updates its spatial belief. We probe this belief by prompting the agent to (i) output a JSON-structured cognitive map of all observed objects and (ii) select unexplored positions from a top-down view with labeled candidate points.

| Methods | Correctness (%) ori. | pos. | overall | Perception (%) ori. | pos. | Local↔Global (%) ori. | pos. | Stability (%) ori. | pos. | Self-tracking (%) ori. | pos. | Uncertainty (%) |
|---|---|---|---|---|---|---|---|---|---|---|---|---|
| | **Vision-based World** | | | | | | | | | | | |
| GPT-5.2 | 20.2 | 42.0 | 32.2 | 33.5 | **72.4** | 57.9 | 58.7 | **65.4** | 56.4 | 93.3 | 64.7 | 53.7 |
| GEMINI-3 PRO | **32.2** | **62.5** | **52.1** | **43.8** | 68.5 | 52.9 | **68.3** | 61.8 | **62.0** | **98.8** | **73.9** | **70.2** |
| | **Text-based World** | | | | | | | | | | | |
| GPT-5.2 | 91.0 | 75.1 | 80.0 | 100 | 86.8 | **96.4** | **86.0** | **96.7** | 67.6 | 98.0 | **86.7** | 64.5 |
| GEMINI-3 PRO | **92.5** | **75.5** | **81.4** | 99.9 | **88.2** | 91.6 | 84.8 | 90.8 | 67.7 | **99.9** | 85.2 | **79.2** |

Table 3: **Spatial Belief Quality via Cognitive Map Probing.** We measure final map correctness and turn-level perception, local global consistency, stability, self-tracking, and uncertainty in text- vs. vision-worlds. *ori.* for orientation and *pos.* for position. Across models, vision lags text on all metrics, with the largest drop on orientation and stability.

beliefs. **Perception** remains a key limitation, particularly for facing direction. This aligns with Table 1, where agents perform poorly on perspective-taking tasks ($\sim 36\%$ accuracy). **Stability & Decay:** While **Perception** scores suggest models can capture local spatial details with reasonable accuracy, this initial fidelity fails to translate into final **Correctness**. Spatial beliefs are highly brittle: even when objects are correctly perceived, agents frequently overwrite verified facts with incorrect predictions in later turns. Low final **Correctness** thus stems not only from perceptual errors, but from the cumulative degradation of spatial memories over the episode.

> 💡 **Key Findings: Cognitive Map Failures (Orientation, Stability, and Belief Drift)**
>
> - **Orientation Gap:** Vision perception is a bottleneck, especially for object orientation.
> - **Unstable Map:** Beliefs about previously observed objects degrades over time.
> - **Belief Drift:** New updates corrupt earlier correct perceptions, lowering final correctness.

We also validate the utility of the probed cognitive map. When prompted to explicitly output the cognitive map before answering, performance degrades slightly, revealing the explicit map as a lossy

| Methods | all | red. | ori. | pos. | ori. | pos. | ori. | pos. |
|---|---|---|---|---|---|---|---|---|
| | Avg. Steps ↓ | | Identification (%) ↑ | | Belief Correctness (%) ↑ | | Belief Inertia (%) ↓ | |
| **Text-based World** | | | | | | | | |
| GPT-5.2 | **6.92** | 0.55 | 97.9 | 98.4 | 89.5 | 69.7 | **5.5** | 12.5 |
| GEMINI-3 PRO | 7.79 | **0.18** | **98.7** | **98.8** | **91.8** | **72.9** | 7.9 | **5.7** |
| **Vision-based World** | | | | | | | | |
| GPT-5.2 | 13.06 | 6.20 | 14.3 | 68.0 | 16.7 | 42.9 | 68.9 | 34.7 |
| GEMINI-3 PRO | **10.29** | **3.23** | **23.9** | **82.5** | **30.3** | **63.1** | **51.1** | **14.4** |

Table 4: **Belief revision results.** We relocate/reorient $k=4$ objects and evaluate change identification, re-exploration cost (including redundancy (red.)), and belief correctness/intertia in text- vs. vision-worlds. Vision agents take more redundant steps and fail to overwrite obsolete facing beliefs.

compression of the model's internal spatial belief. Nonetheless, map **Correctness** is significantly and positively correlated with downstream performance (Pearson $r$ up to $0.645$, all $p < .001$), validating it as a *diagnostic proxy* for failure analysis (Appendix C.4).

> 💡 **Key Findings: Maps as a Diagnostic Proxy**
>
> - **Lossy but Diagnostic:** Though a lossy compression, map correctness correlates significantly with downstream success, making it a strong diagnostic signal.

We additionally include **uncertainty probing**, where the agent identifies previously unobserved locations from a set of candidate points overlaid on an object-free top-down map, evaluated via $F_1$ score. Results show that GEMINI-3 PRO models uncertainty more effectively than GPT-5.2 in both text- and vision-based settings, which aligns with its stronger cognitive map accuracy. Further analysis is provided in Appendix C.3.

## 5.2 BELIEF REVISION TASK

Spatial intelligence requires not only mapping static environments but also maintaining beliefs under non-stationarity. Inspired by *false belief* protocols in developmental psychology (Wimmer & Perner, 1983; Baron-Cohen et al., 1985) and *spatial belief revision* (Knauff et al., 2013), we introduce a dynamic perturbation task to probe the agent's ability to discard obsolete priors and reintegrate new evidence.

**Task Protocol.** Following the initial exploration phase, we introduce a discrete environmental shift: a subset of $k = 4$ objects are stochastically relocated or reoriented. The agent, retaining its memory (exploration history), must actively re-explore the environment to identify the state changes. This requires the agent to detect conflicts between its internal belief state and new sensory observations.

**Metrics.** We evaluate performance along four axes: **Identification Accuracy** ($F_1$) (detect which objects changed position or orientation); **Average Steps** (*Total Steps* to identify all changes, and *Redundancy Steps*, i.e., steps taken after the last changed object is observed, ideally $\rightarrow 0$); **Belief Correctness** (map correctness as in §5.1, restricted to changed objects); and **Belief Inertia** (whether updated beliefs remain systematically biased toward obsolete priors). For inertia, we measure *positional inertia* $s_i^{pos}$ as the tendency of revised beliefs to remain pulled toward their obsolete prior location, and *orientation inertia* $s_i^{ori}$ as the failure rate to overwrite obsolete facing directions. Under unbiased updating, both scores should be near zero. Full definitions are in Appendix A.5.

> 💡 **Key Findings: Vision Deficiencies & Belief Inertia**
>
> - **Vision-based Revision Failures:** Vision agents suffer from excessive exploration redundancy and poor accuracy in identifying object shifts.
> - **Belief Inertia:** Agents, especially vision-based ones, persist in obsolete spatial coordinates despite new observations.

Table 4 corroborates the modality gap observed in previous sections: vision-based agents significantly underperform their text-based counterparts. This performance drop is characterized by increased exploration redundancy and lower accuracy in identifying changed objects. Notably, while belief inertia persists across both modalities, it is markedly more severe in vision-based agents, particularly regarding object orientation. Vision models frequently fail to overwrite their initial spatial memory,

persisting with obsolete facing estimates despite new visual evidence. This also suggests that fine-grained orientation estimation remains a critical bottleneck for visual spatial reasoning.

## 6 RELATED WORK

**Passive Spatial Reasoning.** Early paradigms treat spatial reasoning as static inference: given a textual description, agents answer relational queries (Weston et al., 2015; Shi et al., 2022; Mirzaee et al., 2021; Li et al., 2024). Other benchmarks probe understanding from a single image, asking for relative directions, topological relations, or metric attributes (Ma et al., 2024; Deng et al., 2025; Cheng et al., 2024; Chen et al., 2024; Liao et al., 2024; Kamath et al., 2023). Multi-view and video benchmarks raise difficulty by requiring cross-view integration, egocentric–allocentric conversion, and temporal consistency (Yang et al., 2025c; Xu et al., 2025; Wu et al., 2025; Yeh et al., 2025; Gholami et al., 2025; Zhou et al., 2025b). Recent works explicitly adopt cognitive maps, showing consistent improvements in spatial reasoning across video QA (Yang et al., 2025a) and multi-view settings (Yin et al., 2025). While informative, these benchmarks remain disembodied, as agents reason only over pre-collected trajectories.

**Active Exploration for Spatial Understanding.** Research has also examined agents that actively explore, but their exploration is usually tied to task-specific goals rather than building a general spatial belief. Embodied question answering benchmarks evaluate agents by whether they can gather evidence to answer questions (Das et al., 2018; Gordon et al., 2018; Majumdar et al., 2024; Ginting et al., 2025; Ren et al., 2024). Instruction-following settings extend household tasks to long horizons and realistic scenes, often with dialog or language grounding (Shridhar et al., 2020b; Kim et al., 2024; Shridhar et al., 2020a; Puig et al., 2018; Padmakumar et al., 2022; Gao et al., 2022). Navigation benchmarks stress path execution and generalization across diverse environments (Anderson et al., 2018; Jain et al., 2019; Ku et al., 2020; Krantz et al., 2020; Nguyen & III, 2019; Wang et al., 2024; Zhao et al., 2025). Spatial reference tasks focus on grounding natural-language descriptions in embodied search (Qi et al., 2019; Zhou et al., 2025a), and manipulation (Jiang et al., 2023; Mees et al., 2022; Srivastava et al., 2022; Wu et al., 2023). While existing benchmarks rely on task-driven exploration, they often conflate exploration efficiency with task performance and can foster brittle spatial representations (Bonawitz et al., 2011). EXCALIBUR (Zhu et al., 2023) considers task-agnostic exploration but requires RL training, risking goal leakage and encoding maps implicitly in policy weights. In contrast, we study zero-shot foundation-model agents with no environment-specific training, emphasizing uncertainty reduction over coverage, and evaluating explicit belief construction via belief probing.

## 7 CONCLUSIONS

We introduce THEORY OF SPACE, which asks whether foundation models can function as spatial agents under partial observability: not merely answering questions from fixed views, but actively acquiring information through self-directed exploration to *construct*, *revise*, and *exploit* an internal spatial belief. We contribute an evaluation paradigm centered on task-agnostic active exploration, downstream spatial tasks for belief exploitation, and explicit probing of internal beliefs via cognitive map externalization, implemented in a multimodal environment with parallel text- and vision-based worlds. A key strength of THEORY OF SPACE is that it makes spatial belief *measurable* rather than implicit. By requiring models to externalize evolving cognitive maps, THEORY OF SPACE reveals the correctness, internal consistency, and temporal dynamics of belief formation, and quantifies how localized mistakes propagate into global map corruption. Empirically, **active exploration is a major bottleneck**: end-task performance drops relative to passive viewing, with the gap widening as room complexity increases. Belief probes make error sources explicit: in vision, **perception error** often appears early, and models exhibit belief **instability** where correct information is overwritten or forgotten, cascading into global inconsistencies. When environments change, models exhibit strong **belief inertia**, failing to overwrite obsolete priors. THEORY OF SPACE reframes spatial evaluation from "can the model answer?" to "can the model *build and maintain* a coherent, revisable spatial world model through efficient information gathering?" We hope this benchmark provides a foundation for developing models with uncertainty-aware exploration policies, robust belief maintenance under long horizons, and reliable belief revision when the world changes.

## ETHICS STATEMENT

This work proposes a benchmark for evaluating spatial reasoning in large language and vision-language models. All environments are procedurally generated and rely on publicly available assets without involving human subjects or private data. For human-performance evaluation, all contributors involved in data curation and assessment were compensated at rates well above their local minimum wage. While our benchmark can advance embodied AI research, it may also be applied to domains such as surveillance or autonomous control. We highlight this dual-use potential and encourage responsible application that aligns with ethical guidelines for AI deployment.

## REPRODUCIBILITY STATEMENT

We are committed to ensuring reproducibility of our results. Detailed environment specifications, task definitions, prompts, and examples are included in the main sections §3, §4, and Appendix B. The code and data used in our project is publicly available at https://theory-of-space.github.io/

### ACKNOWLEDGMENTS

This work is in part supported by the Stanford Institute for Human-Centered AI (HAI), ONR YIP N00014-24-1-2117, ONR MURI N00014-22-1-2740, ONR MURI N00014-24-1-2748, ONR MURI N00014-21-1-2801, and DSO National Laboratories Agreement DSOCO25017.

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

APPENDIX CONTENTS

## A   DETAILS OF THEORY OF SPACE

### A.1   FORMAL DEFINITION

> 📄 **Definition: THEORY OF SPACE**
>
> Ability to **construct**, **revise**, and **exploit** an internal spatial belief.

We formalize THEORY OF SPACE within a partially observable framework over a spatial structure $S \in \mathcal{S}$. The agent interacts with $S$ to generate a history $h_t = (o_{0:t}, a_{0:t})$, where $o$ and $a$ denote observations and actions. We define THEORY OF SPACE as the capacity to manipulate a probabilistic belief $B_t$ through three core operations:

1. **Construct:** *To form a globally consistent internal spatial belief by actively seeking out and integrating partial observations.* Formally, the agent integrates $h_t$ to approximate the true posterior, denoted as $B_t(S) \approx P(S \mid h_t)$.

2. **Revise:** *To dynamically update the internal belief by using new information acquired through further exploration to resolve conflicts with prior beliefs.* Upon an environmental shift $S \rightarrow S'$, the agent utilizes exploratory actions $\Delta h$ to minimize the divergence from the new ground truth, i.e., $B_{t+\Delta t} \rightarrow P(S' \mid h_{t+\Delta t})$.

3. **Exploit:** *To utilize the current belief to support spatial tasks.* The agent utilizes a policy $\pi$ conditioned on the belief, $\pi(a_t \mid B_t)$, to perform a downstream task $\mathcal{T}$. In a benchmark context, we measure the *value of belief* by the performance metric $\mathcal{J}$ achieved by this policy: $\mathcal{J}(\pi(\cdot|B_t), \mathcal{T})$.

### A.2   DOWNSTREAM SPATIAL TASKS

Drawing on prior work (Siegel & White, 1975; Montello, 1998), we define tasks to evaluate an agent's **Route** and **Survey** knowledge. An overview of the tasks is present in Figure 4 and Table 5.

### A.3   ASSESSMENT DIMENSIONS

Here we show the detailed definition of five assessment dimensions.

**(D1) Belief Construction Efficiency.** *Measures how efficiently the agent collapses spatial uncertainty during exploration.* We quantify this using a normalized information gain metric, $\mathcal{E}$. Let $M$ be the number of possible positions for any object at the start of exploration (a uniform prior), and let $C_i$ be the number of positions for object $i$ that remain consistent with all observations gathered by the agent (calculated by AC-3 algorithm, check Appendix B.2). The efficiency is calculated as $\mathcal{E} = 1 - \frac{\sum_{i=1}^{N} \log_2 \max(1, C_i)}{N \log_2 M}$. This score ranges from 0 (no information gained, $C_i = M$) to 1 (all objects perfectly localized, $C_i = 1$). Note that it can also be used to calculate the accumulated information gain at each step. Information gain is mainly used in text-based environments, since vision-based environments have direct access to scenes without such ambiguity. Therefore, for vision-based environments, we directly use node coverage to measure exploration efficiency.

**Belief Representation and Quality Assessment.** A core contribution of THEORY OF SPACE is disentangling spatial memory from spatial inference. We structurally decompose the probed cognitive map into two components:

- **(D2) The Cognitive Map (Observed):** *Measures fidelity and coherent integration of observations over time.* We evaluate using two criteria: (1) Correctness, alignment with ground truth, computed as a composite of positional, directional, and facing accuracy; and (2) dynamic reasoning diagnostics, including **Perception** quality, **Self-tracking**, **Stability**, and **Local $\leftrightarrow$ Global Consistency**, reflecting internal coherence such as the absence of contradictions within the relational graph and between maps and relations.

- **(D3) The Uncertainty Map (Unobserved):** *Measures how well the agent models plausible hypotheses about unobserved regions.* We assess **Uncertainty Modeling** by providing a candidate set of positions formed by randomly sampled points from both observed and unobserved areas, and measuring the agent's ability to identify valid locations via $F_1$.

| Dynamic Group | Belief on Route | Belief on Survey |
|---|---|---|
| Static | **Pairwise Relation (*direction*)** report allocentric direction and distance from $A$ to $B$. | **Allocentric Mapping (*alloc.map*)** predict global coordinates (and headings) for all objects. |
| Forward Dynamics | **Perspective Taking (*persp.take*)** output the observation from a specified object's perspective.

**Action-to-View (*act2view*)** given a sequence of `Goto`/`Rotate`, predict the final observation (one object in FOV with ego direction/distance bins). | **Mental Rotation (*ment.rot*)** predict the sequence of front-facing objects during a 360° self-rotation.

**Location2View (*loc2view*)** given a global pose, predict the observation (one object in FOV with ego bins/distances). |
| Backward Dynamics | **Perspective Decision (*perc.dec*)** infer which object's perspective the agent is currently adopting.

**View-to-Action (*view2act*)** recover an action sequence that produces a target observation. | **View2Location (*view2loc*)** localize the agent (and optionally orientation) from a target observation under the map. |

Table 5: **Task suite comparison:** Route belief emphasizes egocentric, step-by-step path reasoning; Survey belief emphasizes allocentric mapping and novel view inference.

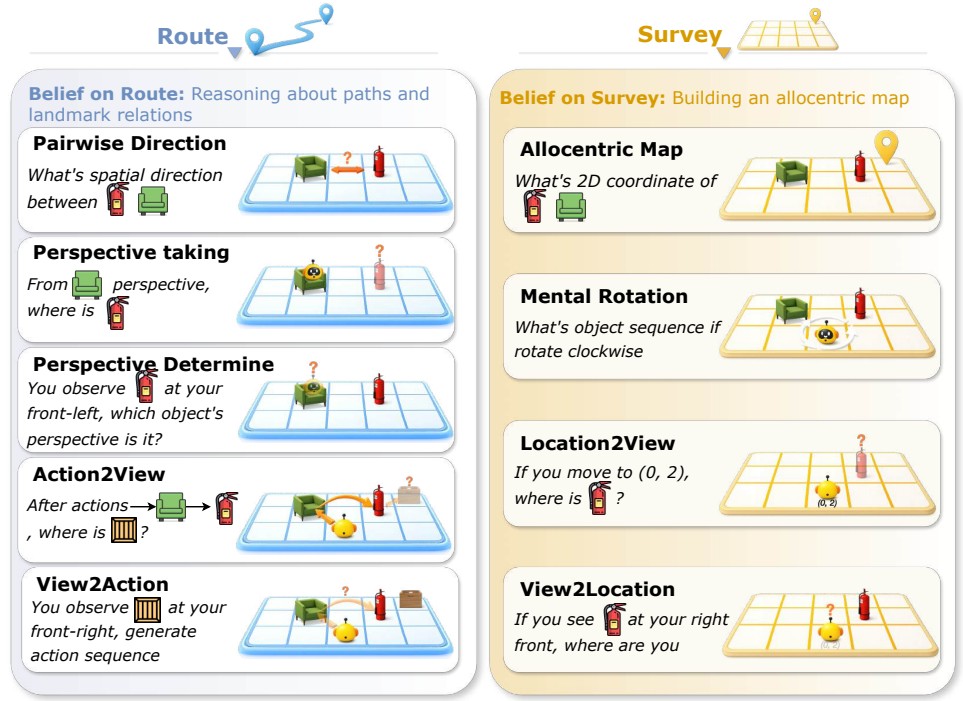

Figure 4: **THEORY OF SPACE exploitation task suite:** it covers **route**-level egocentric reasoning and **survey**-level allocentric mapping. Route tasks evaluate path-based inference and egocentric observations. Survey tasks test global mapping, geometric transformation, and perspective conversion. Together they cover both local navigation reasoning and global spatial abstraction.

This separation lets us diagnose whether failures stem from *misestimating* the observed world or from insufficient *reasoning* about what remains unobserved.

**(D4) Belief Revision.** *Measures the agent's ability to revise its spatial belief under latent environment changes.* We evaluate this using the **False Belief** task (§5.2), where objects are covertly

manipulated (translated or rotated) following the initial exploration. The agent must re-explore to detect these discrepancies; we measure the accuracy of these identified changes (both object identity and transformation type) using the $F_1$ score. Furthermore, we introduce **Belief Inertia** to quantify whether belief revision remain biased toward obsolete priors.

**(D5) Belief Exploitation Success.** *Measures task success when the agent must utilize its spatial belief.* For tasks involving spatial relations (direction, persp.take, action2view), we score direction and distance separately, awarding 0.5 for each correct component. For tasks that output coordinates (view2loc, alloc.map), we compute a coordinate similarity score.

## A.4 COGNITIVE MAP PROBING

We give detailed definition of metrics and scores defined for cognitive map probing here.

**Metrics.** We evaluate cognitive map using three complementary metrics. *Positional accuracy (pos.acc)* is the Euclidean similarity between predicted and true object coordinates: $(K/N) \cdot e^{-\mathrm{RMSE}/L}$, where $\mathrm{RMSE}$ is the root mean squared error between predicted and ground-truth object positions, $L$ is the RMS $\ell_2$-norm of the positions of all objects in the scene, and $K/N$ is the coverage (the ratio of the number of predicted objects K to the number of ground-truth objects N). *Directional accuracy (dir.acc)* is the accuracy of directional relationship between each pair of objects. *Facing accuracy (facing.acc)* is the fraction of objects whose predicted facing matches the ground truth.

Using *global* and *local* belief representations, we compute a set of diagnostic scores at each turn $t$ (all per-turn except **Correctness**, which is computed only at the final turn after termination). Unless noted, scores are averaged over turns and scenes:

- **Correctness (final)**: *Measures the accuracy of the agent's terminal global spatial belief.* At the last turn, we evaluate the predicted global map and report a composite score given by the (equally weighted) mean of the three metrics defined above, with weights $1/3$ each. We compute *dir.acc* only for correctness, since the global cognitive map prioritizes consistent pairwise spatial relations.

- **Perception**: *Measures how accurately the agent interprets newly observed local structure.* We compare the predicted local map to the ground-truth local map for the current field of view (FOV), counting only objects that appear in the FOV for the first time.

- **Self-tracking**: *Measures how well the model estimates its own pose over time.* We infer the agent's pose from the predicted global map and compare it against the ground-truth agent state.

- **Local $\leftrightarrow$ Global consistency**: *Measures whether new local evidence is incorporated into the global belief coherently.* Within the same turn, we compare local and global predictions to verify that newly perceived structure is integrated without contradictions.

- **Stability**: *Measures whether beliefs about previously observed objects remain non-degrading over time.* For each previously observed object, at every subsequent turn we check that its predicted state does not worsen; the per-check score is 1 if the prediction is no worse than in the previous turn.

## A.5 BELIEF REVISION METRICS

We provide detailed definitions of the metrics used to assess belief revision here.

**Metrics.** We evaluate performance along four complementary axes:

- **Identification Accuracy** ($F_1$): *How precisely the agent pinpoints which objects changed.* We compute the $F_1$ score for detecting the subset of objects whose position or orientation shifted.

- **Average Steps:** *How efficiently the agent revises its beliefs to completion.* We report *Total Steps* needed to identify all changes, and *Redundancy Steps*, defined as the number of steps taken after the last changed object has been observed. Ideally, Redundancy $\to 0$, indicating the agent recognizes when updating is complete.

- **Belief Correctness:** *How accurate the updated beliefs are on the changed subset.* We compute correctness as in §5.1, but restrict evaluation to changed objects to isolate the fidelity of re-exploration.

- **Belief Inertia:** *Whether updating remains systematically biased toward obsolete priors.* To quantify attraction back to pre-shift beliefs, we test whether the residual error of the updated belief aligns with the direction of the *old* belief. For each shifted object $i$, let $\mathbf{b}_i^{old}$ denote the pre-shift belief, $\mathbf{b}_i^{new}$ the post-revision belief, and $\mathbf{g}_i^{new}$ the post-shift ground truth. Define the *prior-offset* and *post-revision error* vectors: $\mathbf{v}_i = \mathbf{b}_i^{old} - \mathbf{g}_i^{new}, \mathbf{e}_i = \mathbf{b}_i^{new} - \mathbf{g}_i^{new}$. We define positional inertia as

$$s_i^{pos} = \underbrace{\frac{\mathbf{e}_i^\top \mathbf{v}_i}{\|\mathbf{e}_i\|\,\|\mathbf{v}_i\| + \epsilon}}_{\text{Directional alignment } (\cos\theta_i)} \cdot \underbrace{\exp\left(-\frac{\|\mathbf{b}_i^{new} - \mathbf{b}_i^{old}\|^2}{2\sigma^2}\right)}_{\text{Proximity weight } (w_i)}.$$

Here $\cos\theta_i$ is large when the remaining error after updating still points toward the obsolete location, while $w_i$ downweights such alignment when the belief has moved far from $\mathbf{b}_i^{old}$. We set $\sigma$ to a dynamic noise scale: the RMS localization error on the first re-observed *unchanged* objects during re-exploration; $\epsilon$ ensures numerical stability. Under unbiased updating, $\mathbb{E}[s_i^{pos}] \approx 0$, whereas $s_i^{pos} > 0$ indicates systematic pull toward the obsolete prior. For orientation shifts, we measure inertia via $s_i^{ori} = \mathbb{1}\left(\phi_i^{new} = \phi_i^{old}\right)$, where $\phi$ denotes the predicted orientation. It flags failures to overwrite the obsolete facing direction.

# B TECHNICAL DETAILS

## B.1 BENCHMARK CONSTRUCTION

We expose the ToS world as a *Gym-like* interface (Brockman et al., 2016): agents interact in discrete steps under partial observability at a resolution of $384 \times 384$ to **construct** and **revise** an internal spatial belief, which we later **exploit** in evaluation tasks. Scenes are procedurally generated multi-room layouts on an $N \times M$ grid with $n$ named indoor objects (each with integer $(x, y)$ and heading in {N,E,S,W}) and a randomized agent spawn pose. We restrict multi-room layouts to a tree topology: the room–adjacency graph is connected and acyclic (no loops).

**Text-based World** At each step, OBSERVE returns a symbolic snapshot of objects in the current room within a 90° forward FOV. For every visible object we provide discretized egocentric direction (e.g., *front-left*) and distance bins (e.g., *near/mid/far*), plus object identity and facing when determinable. Egocentric observations are rendered with a 90-degree field of view (FOV), discretized into angular and distance bins as specified in Figure 5a. Visibility is room-bounded; doorways act as transparent portals only when the agent stands in them, enabling dual-room visibility. Optional noise modules perturb bins for ablations.

**Vision-based World** We procedurally generate scenes in a 3D simulator with two controllable parameters: the level (number of rooms) and the object count per room. Objects are drawn from a library of 293 distinct 3D models, grouped into 6 categories and 37 subtypes, primarily everyday household items (see Figure 5b). To ensure diversity, each object type appears at most once in a given scene.

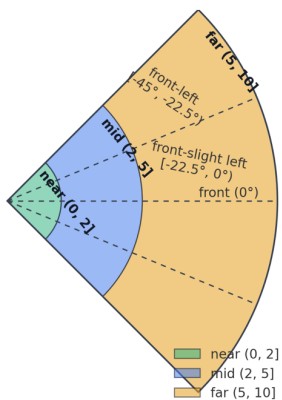
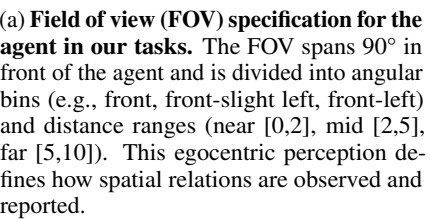

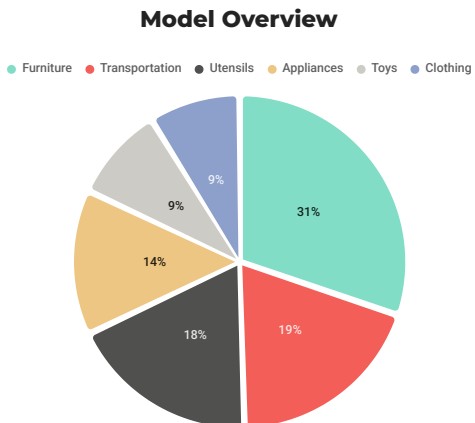

(a) **Field of view (FOV) specification for the agent in our tasks.** The FOV spans 90° in front of the agent and is divided into angular bins (e.g., front, front-slight left, front-left) and distance ranges (near [0,2], mid [2,5], far [5,10]). This egocentric perception defines how spatial relations are observed and reported.

(b) **Distribution of all 3D models used in our vision tasks.**

Figure 5: **Demonstration figures for FOV and 3D model distribution**

For task setup, we additionally generate instructional (Figure 6) and orientation (Figure 7) images that serve as references for the agent in vision-world. We include both images in the vision prompt. Object placement follows validity constraints (e.g., collision avoidance, minimum spacing), and random seeds control reproducibility across environments.

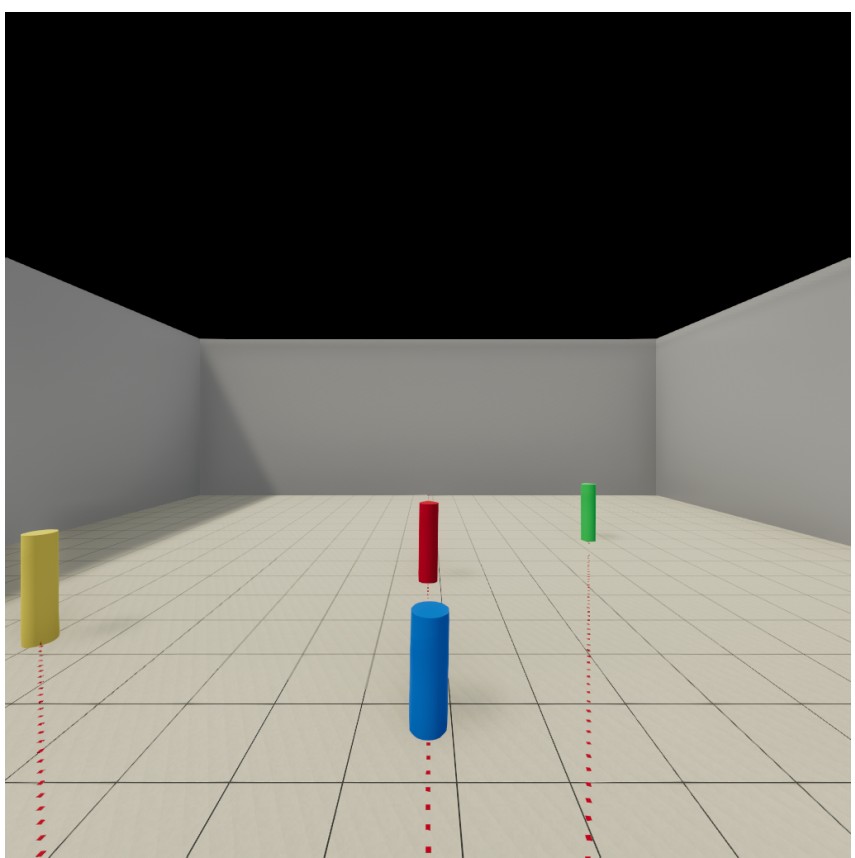

Figure 6: **Example of distance cues in the vision prompt.** The colored cylinders illustrate objects placed at different distances from the agent: yellow at 2 m, blue at 1 m, red at 2 m, and green at 3 m, providing calibration for mapping visual observations to discretized distance bins.

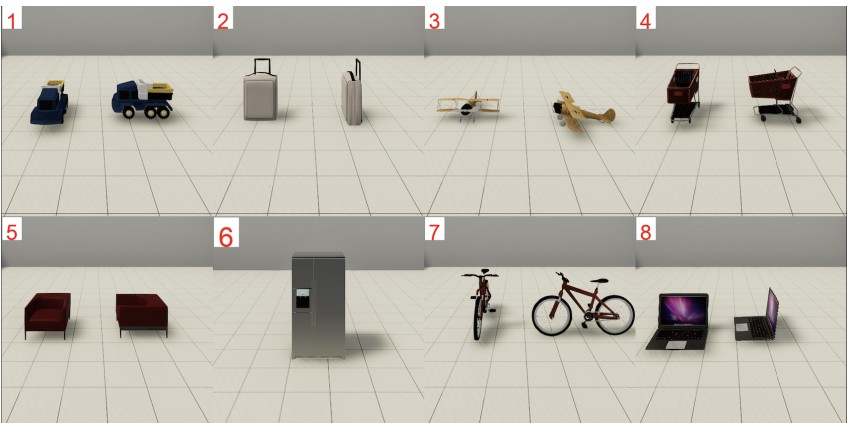

Figure 7: **Object appearance and orientation cues in the vision prompt.** Objects with facing direction are shown from both the front and side views, while objects without inherent orientation are displayed only from the front view. This provides the agent with consistent visual references for recognizing shape and facing.

### B.2    PROXY AGENTS

**Information Gain Calculation**    We use the AC-3 arc-consistency algorithm to maintain, for each object, a domain of feasible grid cells. Initially, every object's domain spans the entire $20 \times 20$ map.

Each new observation is compiled into unary and binary constraints (e.g., egocentric direction/distance bins, room visibility/occlusion, and ALLDIFFERENT to prevent collisions). When a constraint is added, AC-3 iteratively prunes any cell in one object's domain that is unsupported by the domains of related objects, propagating revisions along incident arcs until a fixed point is reached (all arcs are consistent). While AC-3 alone does not guarantee global consistency, in our setting all constraints are derived from a valid trajectory; therefore the ground-truth assignment remains supported and is never pruned, ensuring that domains stay non-empty throughout propagation.

**Scripted Proxy Agents** We implement two scripted proxies to provide strong, reproducible baselines.

SCOUT. From its spawn pose, the agent performs a 360° sweep (four cardinal ROTATE+OBSERVE actions) to capture all views at the initial location. It then follows a fixed room-visitation order: upon discovering a doorway, it enters the adjacent room, executes the same sequential sweep, and repeats this "visit–sweep–advance" routine until every room has been observed at least once.

STRATEGIST. The first stage mirrors SCOUT: a panoramic sweep to register all currently visible objects. Thereafter, within the *current room* the agent maintains, for each object, a set of feasible positions ("domain") induced by accumulated observations. At each turn it: (i) selects the object with the largest remaining domain (highest positional uncertainty); (ii) moves to a viewpoint that best constrains this object (e.g., near it or along a sightline that intersects the most candidate cells); (iii) at that viewpoint, orients to test pairwise relations: it computes unresolved pairwise directions between the target object and all others in the room, identifies the direction bin with the highest outstanding count, and OBSERVEs in that orientation first. The procedure iterates until all objects in the room are resolved (domains shrink to singletons), then proceeds to the next unvisited room and repeats.

## B.3 PROMPTS

We show the detailed designs of our prompts for exploration in Figure 8, evaluation prompts in Figure 9, cognitive map prompts in Figure 10, and top-down view for uncertainty modeling in Figure 11.

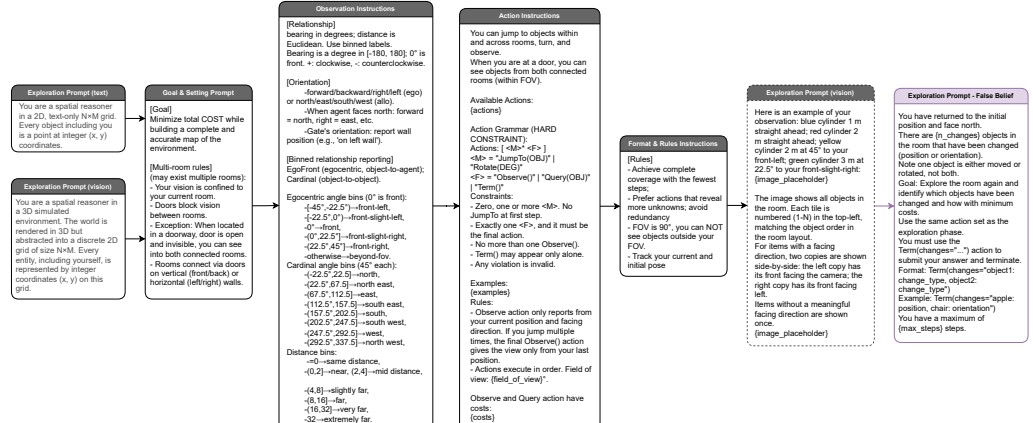

Figure 8: **Exploration prompts**

## Route

**Pairwise Direction**

You return to your starting position and face north. From a Top-Down map, describe where {obj_name} is relative to {anchor_name}.

Answer format: <cardinal direction>, <distance>

Example: north-west, near

**Perspective Taking**

Now you jump to {anchor_name}'s direction, facing its direction. Describe where {obj_name} is relative to you.

Answer format: <ego direction>, <distance>

Example: front-left, near

**Perspective Determine**

Now you jump to an object's position, facing its direction. You observe that {observation}. Which object are you standing at?

Answer format: <object_name>

Example: lamp

**Action2View**

You return to your starting position and face north. You will execute the following action sequence: {actions}

After executing the actions, what is the ego relation of {target} relative to you?

Answer format: <ego direction>, <distance>
Example: front, near

**View2Action**

You return to your starting position and face north. Then you have executed an action sequence and changed to a new location and facing direction. You observe the following: {final_obs}

What action sequence led to this final view? The action sequence must be valid and only contain move actions.

Answer format: <sequence of move actions>
Example: JumpTo(lamp), Rotate(90)

## Survey

**Allocentric Map**

Treat your starting position as the origin (0, 0) while facing north. Report allocentric coordinates using (x right/east, y up/north). Objects: {object_list}.

Answer format: (x0, y0); (x1, y1); ... in the same order.

Example: (1, 0); (-2, 3); (0, -1)

**Mental Rotation**

You return to your starting position and face north. You will perform a full 360-degree rotation by continuously turning {turn_direction} in place. Assume all walls are removed (you can see through walls), so every object is visible.

Focus on this set of objects: {object_pool}.
List them in the exact order they appear directly ahead while you rotate.
If two objects share a bearing, place the nearer one first.

Answer format: <object_name1>, <object_name2>, ...

Example: mug, sofa, plant

**Location2View**

{origin_instruction}You move to {loc} and face {direction}.
What is the egocentric relation of {target}?

Answer format: <direction>, <distance>

Example: front, near

**View2Location**

You move to a new location and face {orientation}. {observations}
{origin_instruction}What is your new 2D coordinate (x, y)?

Answer format: (x, y)
Example: (2, -1)

Figure 9: **Evaluation prompt design. We show the prompt for each evaluation task.**

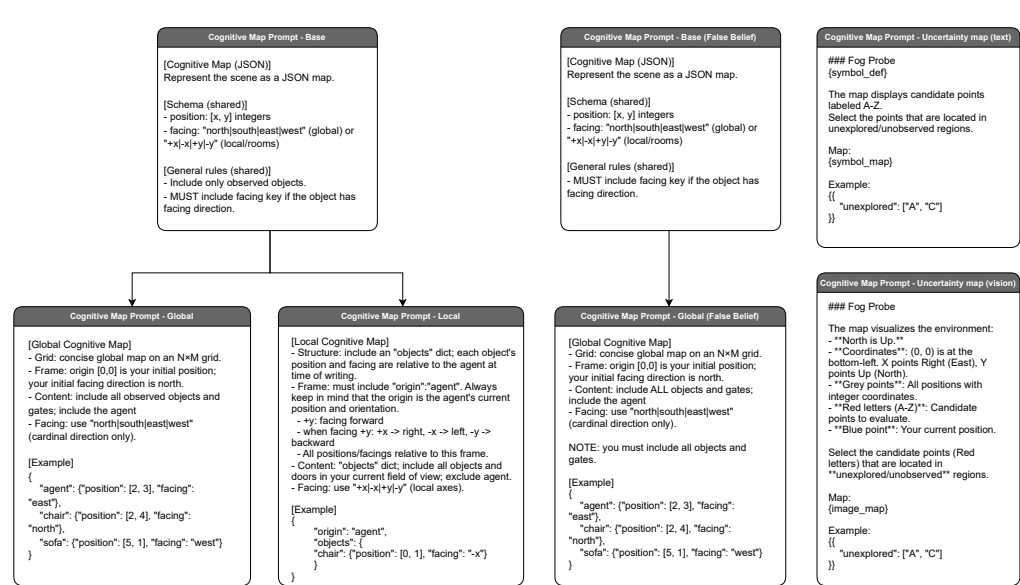

Figure 10: **Belief probing prompt design. We use these prompts to ask the model to output a cognitive map or select unobserved points.**

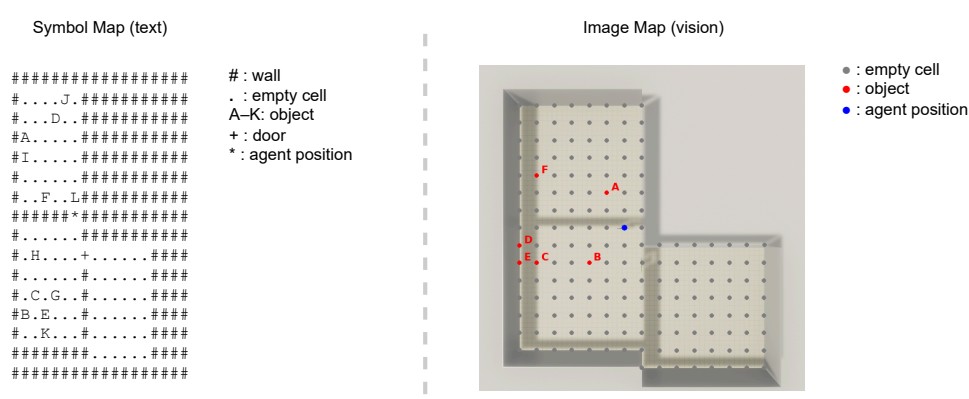

Figure 11: **The symbol map and the image map** provide parallel representations of the same environment for text and vision settings in uncertainty probing prompts.

## C    EXTRA RESULTS

To enable a like-for-like comparison between the text and vision settings, we instantiate identical room layouts across modalities. Concretely, we generate 100 evaluation instances with IDs 0–99; for each ID, we use the ID itself as the random seed to drive task sampling in both environments. This seed tying guarantees deterministic layouts and bit-for-bit reproducibility across modalities.

### C.1    ADDITIONAL RESULTS FOR PASSIVE SETTING

The performance of various models in both text-based and vision-based environments is summarized in Table 6.

| Methods | direction | persp.take | perc.dec | act2view | view2act | alloc.map | ment.rot | loc2view | view2loc | Avg. |
|---|---|---|---|---|---|---|---|---|---|---|
| | *Static (S)* | | *Dynamic (D)* | | | *Static (S)* | | *Dynamic (D)* | | |
| | **Route** | | | | | **Survey** | | | | |
| **Vision-based World** | | | | | | | | | | |
| *Proprietary Models* | | | | | | | | | | |
| GPT-5.2 | 47.3 | 35.0 | **63.9** | **54.5** | 49.3 | 64.8 | 83.3 | 50.3 | **65.6** | 57.1 |
| GEMINI-3 PRO | **63.8** | **36.3** | 57.5 | 49.0 | **58.0** | 67.2 | 85.3 | 70.4 | 57.0 | **60.5** |
| CLAUDE-4.5 SONNET | 47.3 | 33.5 | 37.7 | 40.8 | 15.7 | 54.8 | 58.3 | 44.7 | 54.8 | 43.1 |
| *Open-source Models* | | | | | | | | | | |
| GLM-4.6V | 11.5 | 24.5 | 4.7 | **19.0** | 2.7 | 22.9 | 11.7 | 20.0 | 33.6 | 16.7 |
| QWEN3-VL | **20.8** | **28.3** | **22.7** | 16.7 | **4.7** | **33.2** | **21.7** | **27.3** | **40.8** | **24.9** |
| **Text-based World** | | | | | | | | | | |
| *Proprietary Models* | | | | | | | | | | |
| GPT-5.2 | **84.5** | 88.2 | **97.0** | **89.0** | **76.0** | **96.3** | **98.3** | **94.8** | **89.2** | **90.4** |
| GEMINI-3 PRO | 82.7 | **92.7** | **97.0** | 87.5 | 75.7 | 86.2 | 91.3 | 85.7 | 80.0 | 86.5 |
| CLAUDE-4.5 SONNET | 73.0 | 80.7 | 90.7 | 77.7 | 59.0 | 76.9 | 74.3 | 59.2 | 70.7 | 73.6 |
| *Open-source Models* | | | | | | | | | | |
| GLM-4.6V | 22.3 | 39.8 | 25.0 | 25.3 | 4.7 | 21.2 | 9.0 | 27.0 | 35.7 | 23.4 |
| INTERNVL-3.5 | 36.7 | 67.8 | 42.7 | 41.2 | 8.7 | 37.3 | 19.3 | 38.7 | 43.8 | 37.4 |
| QWEN3-VL | **40.8** | **69.3** | **56.5** | **50.0** | **17.7** | **42.8** | **40.3** | **42.5** | **54.6** | **45.6** |

Table 6: **Exploitation Performance** (%) **of Belief Construction via Passive Observations.** Models are evaluated as *passive comprehension agents* on Route- and Survey-level reasoning using standard-ized observation logs from scripted proxy explorers, decoupling exploration from belief construction across text- and vision-based environments. GEMINI-3 PRO leads most tasks in the vision-based world and achieves the best overall average, while GPT-5.2 leads the text-based world and attains the best overall average.

### C.2    ADDITIONAL RESULTS FOR MULTI-ROOM SETTINGS

We show detailed results for **different room settings** including two-room and four-room layouts. In both the two-room and four-room settings, we use the same room size and the same number of objects per room as in the three-room setting. For the four-room setting, we connect the main room with all the others. We evaluate GPT-5.2 and GEMINI-3 PRO, the two best-performing models. **Additionally, we tested higher resolution, but found no performance gain.** Table 7 and 8 report passive and active performance of the two-room setting. Table 9 and 10 report passive and active performance of the three-room setting. As the number of rooms increases, exploration cost rises accordingly. The results also underscore the importance of efficient exploration: in the four-room setting, which demands more strategic exploration, the gap between active and passive performance becomes substantially larger.

| Methods | direction | persp.take | perc.dec | act2view | view2act | alloc.map | ment.rot | loc2view | view2loc | Avg. |
|---|---|---|---|---|---|---|---|---|---|---|
| | Static (S) | Dynamic (D) | | | | Static (S) | Dynamic (D) | | | |
| | Route | | | | | Survey | | | | |
| *Vision-based World* | | | | | | | | | | |
| *Proprietary Models* | | | | | | | | | | |
| GPT-5.2 | 39.2 | 37.3 | 63.3 | 53.8 | 58.3 | 68.2 | 92.7 | 52.3 | 68.6 | 59.3 |
| GEMINI-3 PRO | 57.8 | 33.9 | 53.8 | 48.5 | 58.7 | 64.6 | 83.3 | 54.7 | 69.8 | 58.3 |
| *Text-based World* | | | | | | | | | | |
| *Proprietary Models* | | | | | | | | | | |
| GPT-5.2 | 85.3 | 92.0 | 99.0 | 90.0 | 83.0 | 97.2 | 99.7 | 89.5 | 95.2 | 92.3 |
| GEMINI-3 PRO | 88.2 | 86.7 | 91.7 | 87.3 | 79.3 | 90.1 | 92.7 | 81.5 | 82.9 | 86.7 |

Table 7: Exploitation Performance (%) via Passive Observations under **two rooms** settings.

| Methods | Avg.cost | direction | persp.take | perc.dec. | act2view | view2act | alloc.map | ment.rot | loc2view | view2loc | Avg. |
|---|---|---|---|---|---|---|---|---|---|---|---|
| | | Static (S) | Dynamic (D) | | | | Static (S) | Dynamic (D) | | | |
| | | Route | | | | | Survey | | | | |
| *Vision-based World* | | | | | | | | | | | |
| *Proprietary Models* | | | | | | | | | | | |
| GPT-5.2 | 10.8 | 41.3 | 36.2 | 48.2 | 49.0 | 54.7 | 56.9 | 72.0 | 45.2 | 59.7 | 51.5 |
| GEMINI-3 PRO | 6.6 | 51.7 | 36.3 | 63.0 | 47.2 | 56.0 | 63.4 | 85.0 | 50.3 | 67.5 | 57.8 |
| *Text-based World* | | | | | | | | | | | |
| *Proprietary Models* | | | | | | | | | | | |
| GPT-5.2 | 6.2 | 68.7 | 67.3 | 90.0 | 76.8 | 64.0 | 83.4 | 92.7 | 73.7 | 83.7 | 77.8 |
| GEMINI-3 PRO | 6.2 | 76.0 | 68.3 | 89.0 | 77.2 | 72.7 | 83.1 | 96.0 | 77.5 | 86.2 | 80.6 |

Table 8: Exploitation Performance (%) via Active Exploration under **two rooms** settings.

| Methods | direction | persp.take | perc.dec | act2view | view2act | alloc.map | ment.rot | loc2view | view2loc | Avg. |
|---|---|---|---|---|---|---|---|---|---|---|
| | Static (S) | Dynamic (D) | | | | Static (S) | Dynamic (D) | | | |
| | Route | | | | | Survey | | | | |
| *Vision-based World* | | | | | | | | | | |
| *Proprietary Models* | | | | | | | | | | |
| GPT-5.2 | 47.0 | 37.7 | 59.7 | 38.3 | 40.3 | 60.1 | 73.7 | 50.5 | 65.9 | 52.6 |
| GEMINI-3 PRO | 63.5 | 35.5 | 58.7 | 42.8 | 43.0 | 64.4 | 81.7 | 48.8 | 67.4 | 56.2 |
| *Text-based World* | | | | | | | | | | |
| *Proprietary Models* | | | | | | | | | | |
| GPT-5.2 | 83.8 | 88.2 | 94.3 | 86.8 | 62.7 | 94.8 | 93.7 | 82.0 | 92.5 | 86.5 |
| GEMINI-3 PRO | 81.2 | 91.3 | 96.7 | 82.2 | 68.3 | 76.8 | 81.3 | 74.2 | 79.0 | 81.2 |

Table 9: Exploitation Performance (%) via Passive Observations under **four rooms** settings.

| Methods | Avg.cost | direction | persp.take | perc.dec. | act2view | view2act | alloc.map | ment.rot | loc2view | view2loc | Avg. |
|---|---|---|---|---|---|---|---|---|---|---|---|
| | | Static (S) | Dynamic (D) | | | | Static (S) | Dynamic (D) | | | |
| | | Route | | | | | Survey | | | | |
| *Vision-based World* | | | | | | | | | | | |
| *Proprietary Models* | | | | | | | | | | | |
| GPT-5.2 | 23.2 | 41.2 | 33.2 | 49.0 | 30.8 | 30.7 | 32.5 | 49.7 | 40.5 | 55.4 | 40.3 |
| GEMINI-3 PRO | 19.7 | 59.8 | 34.2 | 60.3 | 34.7 | 46.0 | 56.8 | 62.7 | 44.0 | 64.8 | 51.5 |
| *Text-based World* | | | | | | | | | | | |
| *Proprietary Models* | | | | | | | | | | | |
| GPT-5.2 | 16.4 | 65.3 | 69.0 | 74.3 | 62.8 | 44.3 | 66.6 | 76.3 | 57.5 | 77.8 | 66.0 |
| GEMINI-3 PRO | 19.7 | 76.3 | 77.2 | 91.7 | 73.3 | 64.3 | 77.0 | 83.7 | 74.0 | 81.9 | 77.7 |

Table 10: Exploitation Performance (%) via Active Exploration under **four rooms** settings.

## C.3 UNCERTAINTY MAP PROBING

To probe an agent's ability to model uncertainty, we provide it with a top-down view of the scene in which all objects are removed, and we overlay a set of candidate points. These points are sampled randomly and include both previously observed and unobserved locations. The agent's task is to identify which candidate points remain unobserved, thereby revealing its belief over unseen regions.

**Representation.** The agent receives an empty top down map that shows only the candidate points and its current position, with no objects present. The agent must select the points that have not yet been observed. In the text based world, the top down map is represented as an $N \times M$ symbolic grid, where different symbols denote the agent, gates, and candidate points. In the vision based world,

Figure 12: Accumulated Information Gain and Cognitive Map Correctness over steps.

all objects are removed and the agent instead receives a top down image of the environment, check examples in Figure 11. We use $F_1$ to evaluate selected points.

We report **Uncertainty** scores in Table 3. GEMINI-3 PRO models uncertainty better than GPT-5.2 in both text- and vision-based settings. These results help explain the information gain and cognitive map trends in Figure 12. GPT-5.2 achieves higher initial information gain (i.e., it ramps up faster), likely because it quickly commits to an explore-the-doors strategy. However, it generalizes poorly to unobserved regions, reflected by the subsequent plateau in Figure 12: additional steps yield little marginal gain. In contrast, although GEMINI-3 PRO improves more slowly at the beginning, its cognitive map accuracy continues to increase with exploration, suggesting it keeps collecting useful evidence and progressively resolving uncertainty.

## C.4 COGNITIVE MAP VALIDATION & CORRELATION

**Cognitive Map Validation & Correlation.** To validate the utility of the probed cognitive map and investigate whether it faithfully reflects the agent's reasoning process, we first conducted two ablation studies:

- **Sufficiency Test (Oracle Map):** We conditioned the model on the ground-truth cognitive map before generating answers for evaluation. Performance rose to near-perfect levels ($\approx 95\%$ for both models in both worlds). This confirms that our cognitive map representation captures *all* necessary information for the tasks; performance bottlenecks stem from the agent's inability to accurately *construct* the map, not the representation format itself.

- **Alignment Test (Explicit Reasoning):** We prompted the model to explicitly generate the cognitive map before answering the evaluation questions. This resulted in a slight performance degradation compared to direct answering.

These results reveal an **externalization gap**: the model's latent internal spatial belief is richer or more accurate than the discretized JSON output it produces. **While it is a lossy compression of the agent's true internal state, the explicit map remains a strong diagnostic signal.** We support this claim by computing the Pearson correlation between the agent's cognitive map **Correctness** and downstream task performance. To ensure a robust correlation, we calculate the average performance across five independent cognitive map runs for each sample. As shown in Table 11, belief correctness is consistently and positively correlated with

| Methods | Text (%) | Vision (%) |
|---|---|---|
| GPT-5.2 | 41.8 | 57.0 |
| GEMINI-3 PRO | 46.6 | 64.5 |

Table 11: **Pearson correlation** ($r$) between belief correctness and downstream performance. All correlations are significant ($p < .001$).

downstream success in both modalities, with all correlations significant ($p < .001$). The association

is stronger in vision ($r{=}0.570/0.645$) than in text ($r{=}0.418/0.466$). The stronger vision correlation suggests that perception-driven mapping errors and unstable belief updates more directly translate into task failures. Thus, we establish map probing as a *validated diagnostic proxy* for failure analysis. While acknowledging that correlation does not imply causality, we treat the explicit map as a robust, albeit conservative, signal for diagnosing reasoning breakdowns rather than definitive evidence.

# D    ADDITIONAL VISUALIZATION EXAMPLES

We include concrete examples of task formats and answer styles with open-ended, format-constrained outputs in Figure 13.

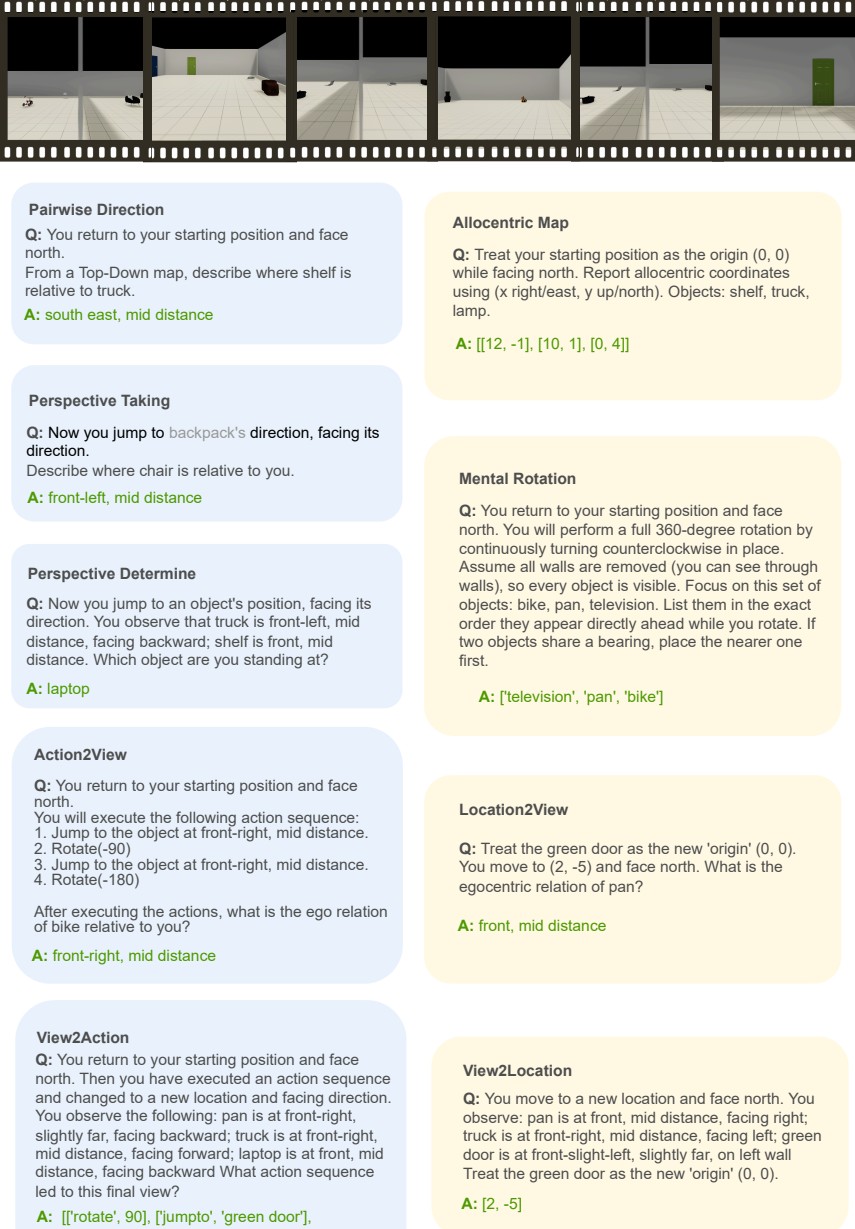

Figure 13: **Examples of task formats and answer styles used.** Each block illustrates a spatial reasoning task type in our suite (Route-level and Survey-level), including the corresponding input context and an example open-ended answer that must follow a strict output format. In the vision setting, textual scene descriptions in the questions are replaced by rendered observation images.

## D.1 COGNITIVE MAP OUTPUT BY MODELS

We visualize the turn-by-turn cognitive maps (in Figures 14 and 15 of GPT-5.2, comparing them against ground-truth maps. The performance is noticeably stronger in text-based environments than in vision-based ones.

## D.2 EXPLORATION PATTERN EXAMPLES BY MODELS

We include representative trajectories from each model to illustrate the active exploration patterns identified in our analysis, shown in Figure 16, 17, 18, 19, and 20 . These examples highlight how different models manifest recurring exploration behaviors: for instance, GPT-5.2 often adopts a "finding-gate" strategy, rotating until a doorway is detected before moving toward it, while other models more frequently repeat redundant checks. All figures mark the agent's position and orientation explicitly, with actions annotated beneath each frame and a shared legend provided for each trajectory.

## D.3 ANALYSIS PLATFORM

We also include some demonstrations in Figure 21, 23, 22, 24, and 25 of our designed platform for better analysis

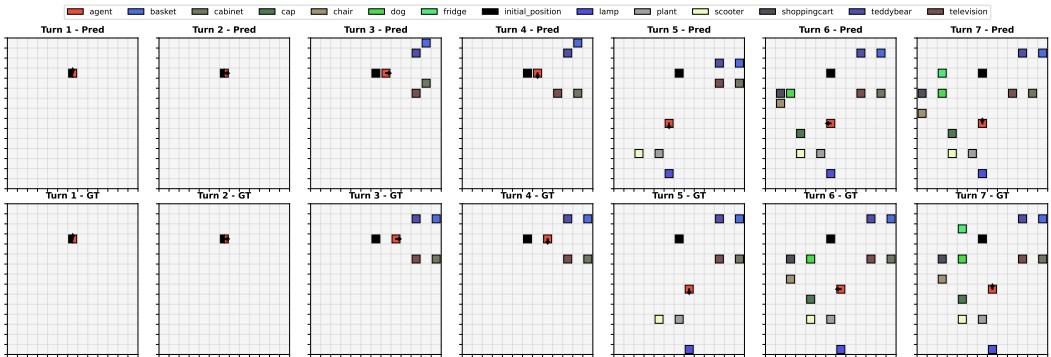

Figure 14: GPT-5.2's turn-by-turn cognitive map in text world during exploration.

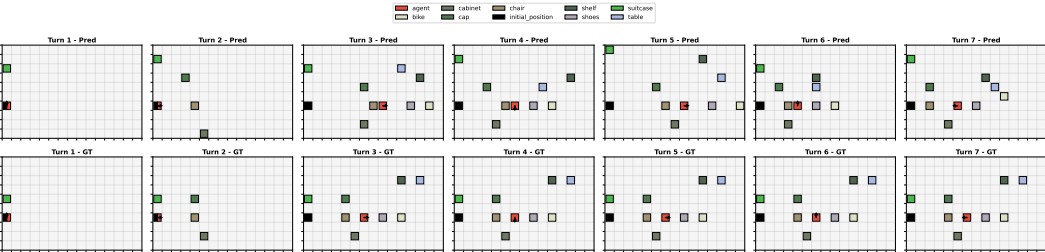

Figure 15: GPT-5.2's turn-by-turn cognitive map in vision world during exploration.

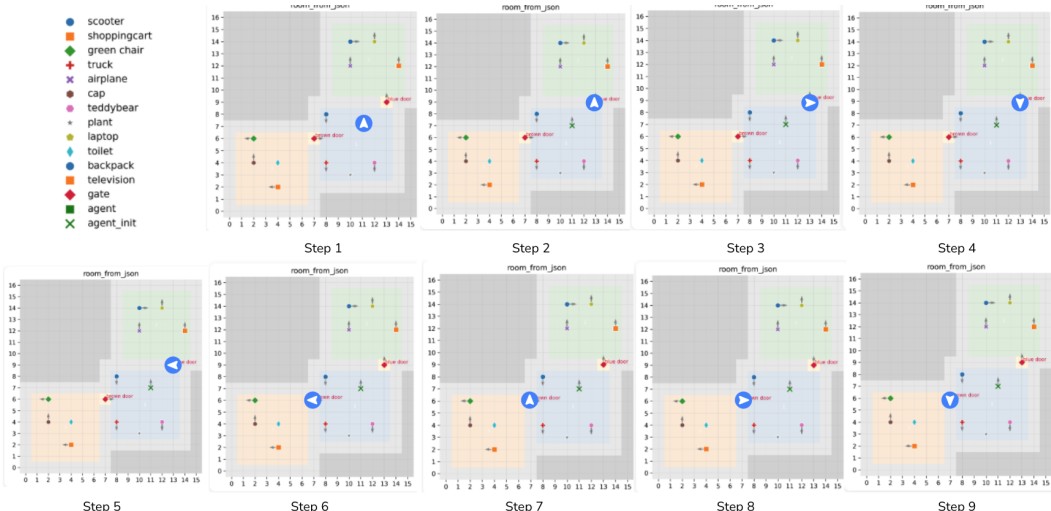

Figure 16: Example trajectory illustrating GPT-5.2's door-finding strategy and systematic sweeping pattern: Upon detecting a door, the agent navigates toward it and executes a strategic rotation to maximize environmental coverage. The process terminates once all target objects have been successfully identified.

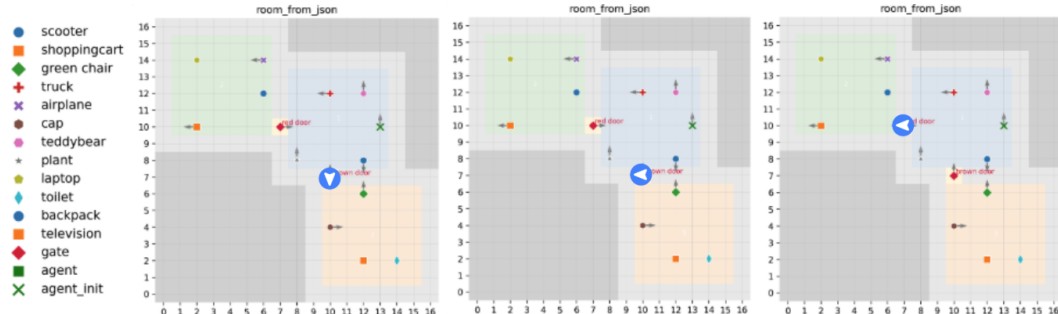

Figure 17: Example trajectory illustrating GPT-5.2's omission pattern: Observing the door too early may lead the agent to skip the rest of the exploration, causing incomplete environmental discovery.

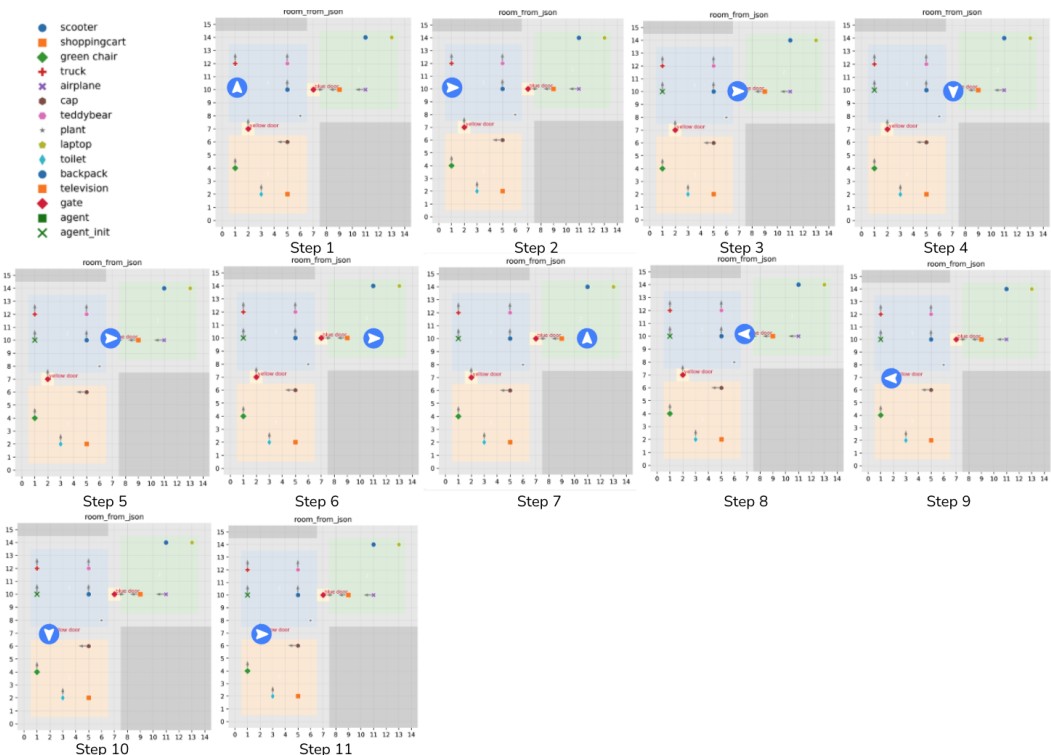

Figure 18: Example trajectory illustrating GEMINI-3 PRO's door-finding strategy and systematic sweeping pattern in vision world: Upon detecting a door, the agent navigates toward it and executes a strategic rotation to maximize environmental coverage. The process terminates once all target objects have been successfully identified.

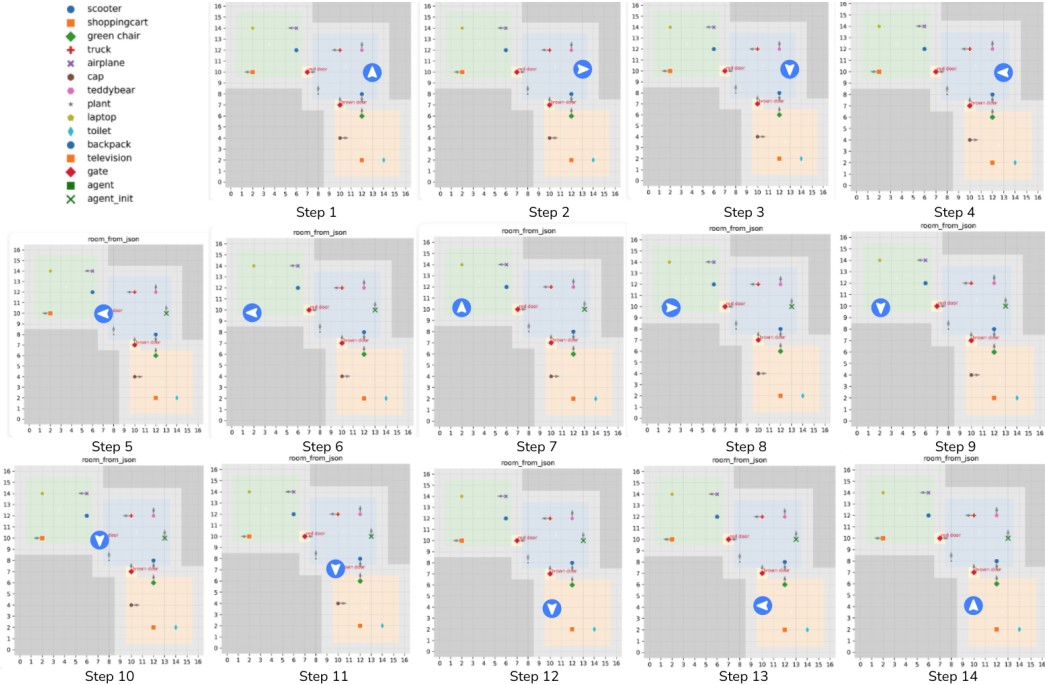

Figure 19: Example trajectory illustrating GEMINI-3 PRO's object sweeping pattern mostly found in text world: Orbit the starting object using it as the pivot point. Randomly select an observed door to jump to a new object, then resume pivoting around the new target in a continuous loop.

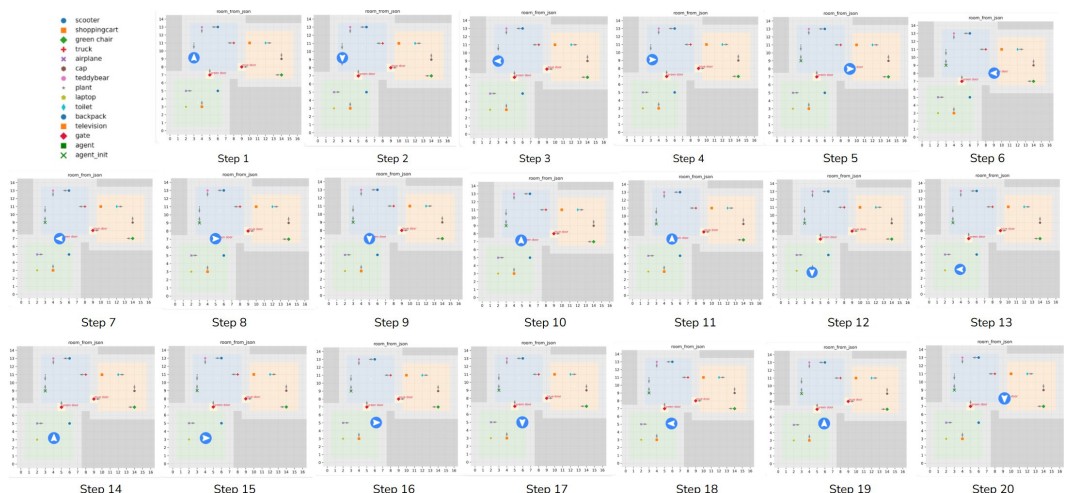

Figure 20: Example trajectory illustrating CLAUDE-4.5 SONNET's exploration pattern: There is no clear exploration pattern.

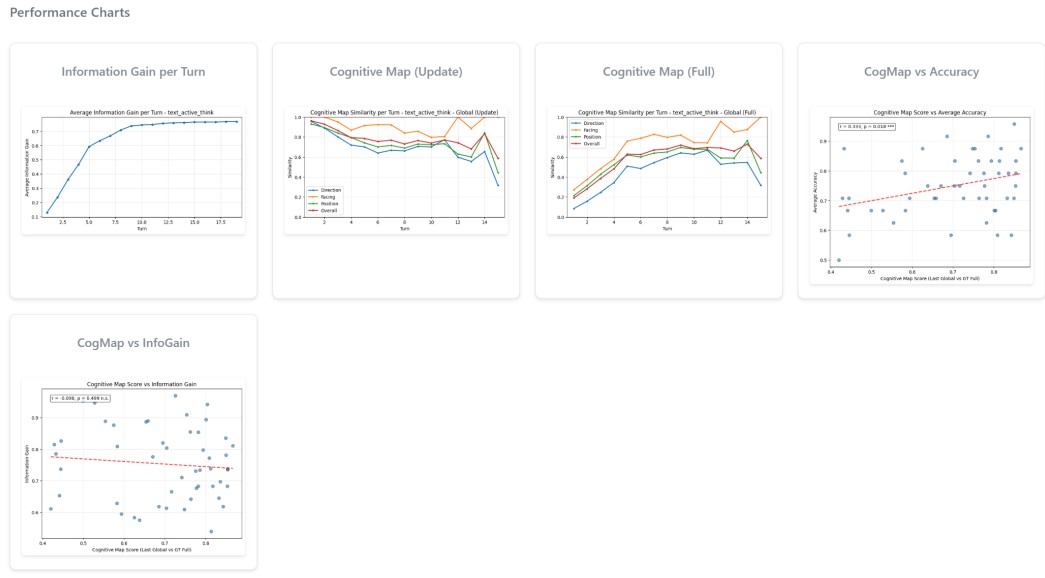

Figure 21: Platform designed by us for analysis (chart)

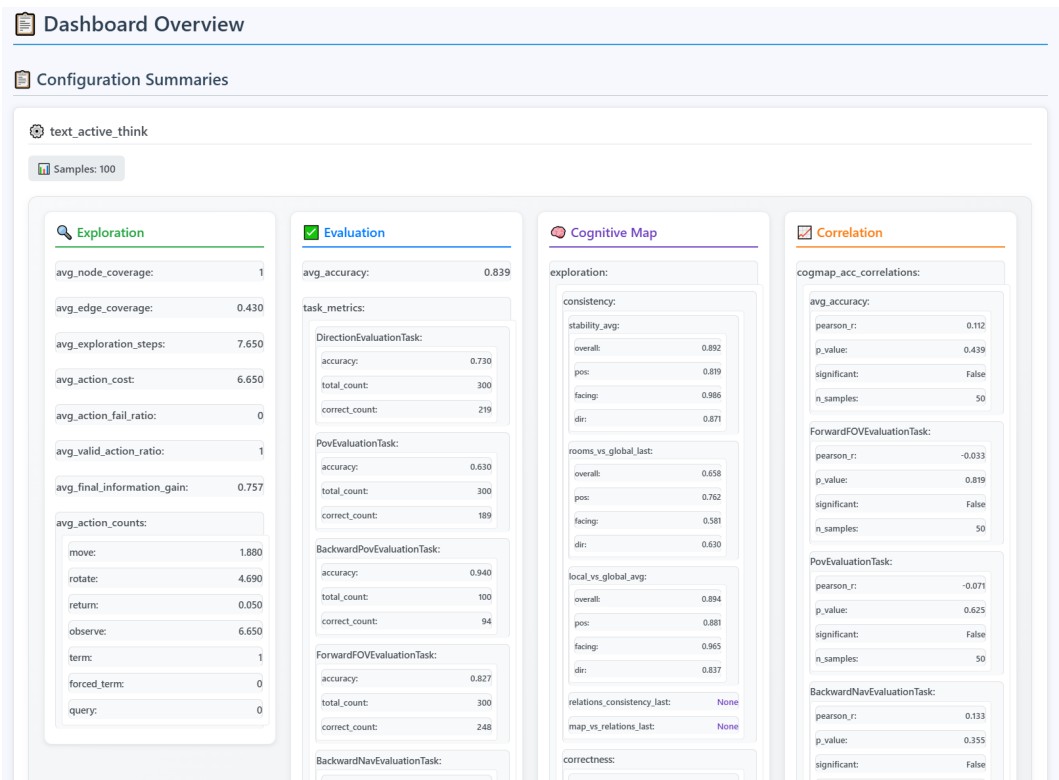

Figure 22: Visualization Platform for analysis: Metrics for active exploration in text world

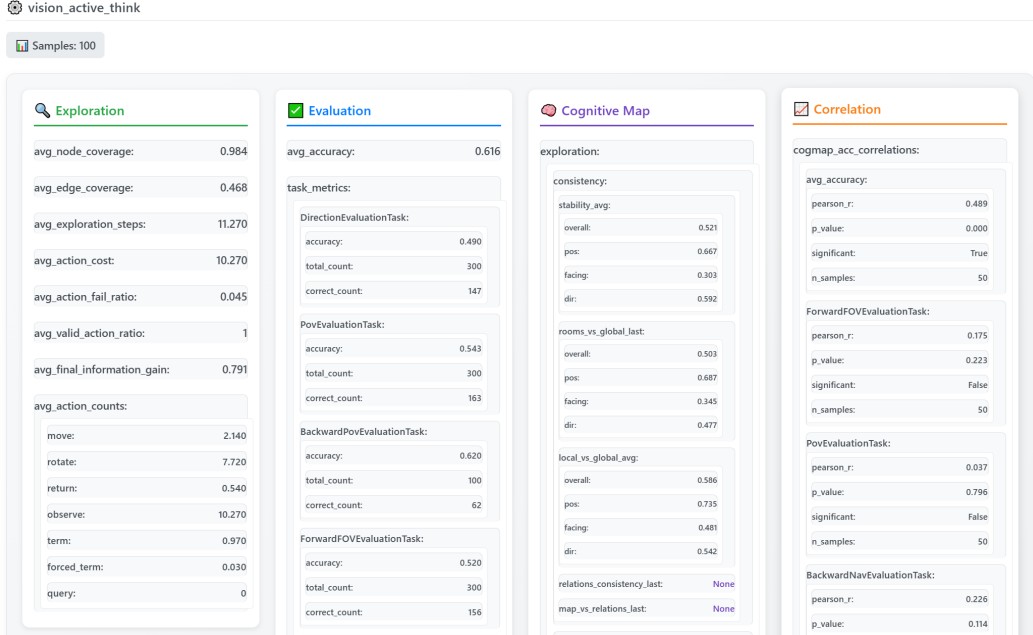

Figure 23: Visualization Platform for analysis: Metrics for active exploration in vision world

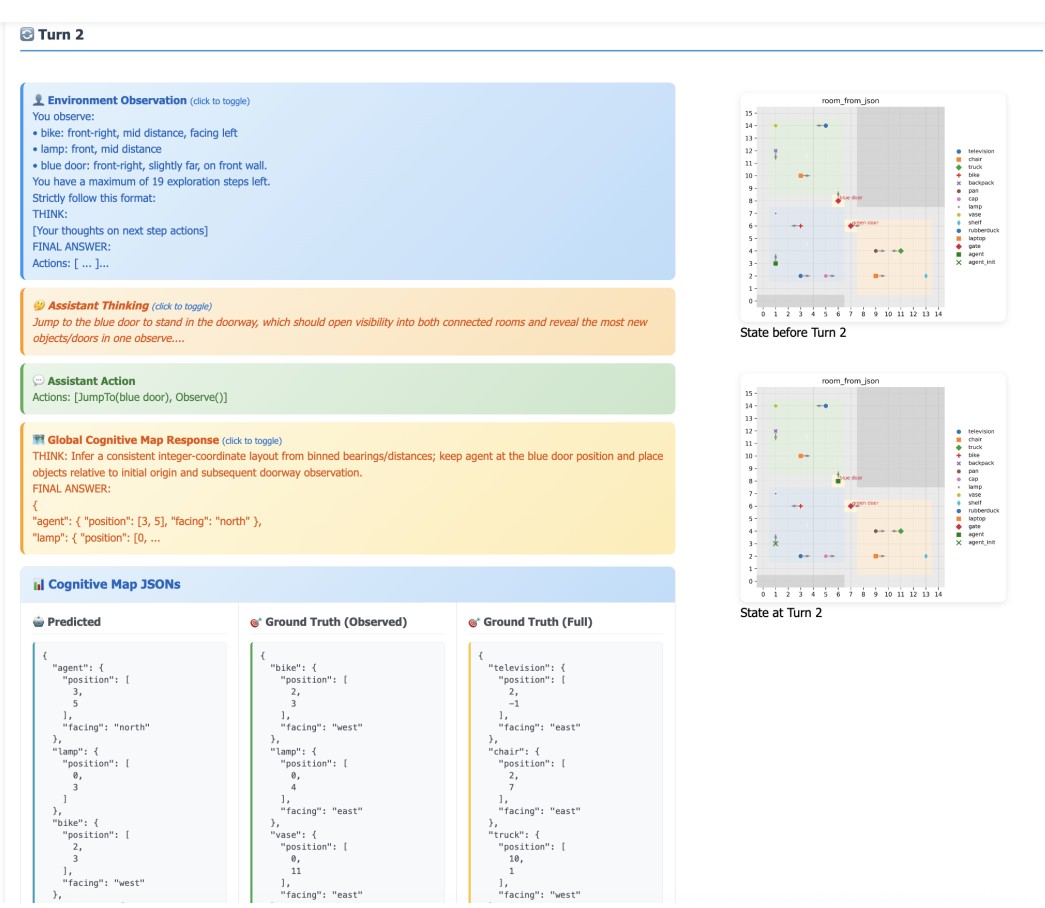

Figure 24: Visualization Platform for analysis: one turn of active exploration in text-world, including agent's action and cognitive map.

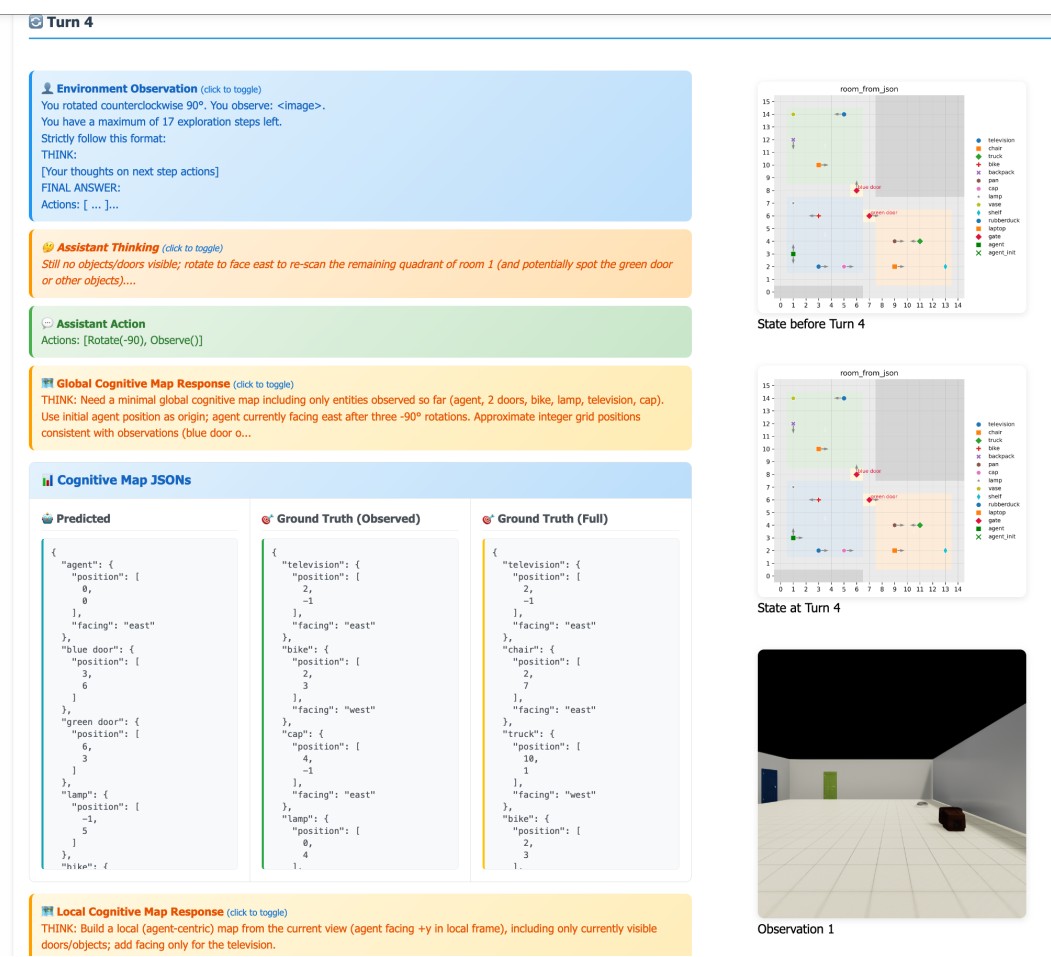

Figure 25: Visualization Platform for analysis: one turn of active exploration in vision-world

# E   LLM USAGE STATEMENT

We used Large Language Model like GEMINI-3 PRO and GPT-5.2 as a writing assistant such as improving readability and phrasing. LLMs did not contribute novel content, proofs, or results. All research ideas, analyses, experiments, and results were developed by the authors, who take full responsibility for the paper's accuracy and integrity.

