# OpenReview forum: "Theory of Space: Can Foundation Models Construct Spatial Beliefs through Active Exploration?"
_ICLR.cc/2026/Conference — ICLR 2026 Poster_

### Official Review · Reviewer_TnFX · 2025-10-21

**Soundness:** 3
**Presentation:** 4
**Contribution:** 3
**Rating:** 6
**Confidence:** 3

**Summary:**

This paper argues that today’s VLM agents excel at passive or task-driven spatial reasoning but largely fail to build a coherent understanding of space through exploration. It proposes “Theory of Space” (ToS): an agent’s ability to construct, update, and use an internal spatial belief from partial, egocentric observations, and introduces a benchmark that makes agents actively explore text and vision environments (procedural multi-room layouts in a grid world and a ThreeDWorld+Objaverse simulator).

 At every step the agent must externalize its belief as a cognitive map (JSON), which affords assessing not just task accuracy but belief correctness, consistency, and stability. The suite spans route and survey tasks (e.g., pairwise relations, action-to-view, allocentric mapping, mental rotation, localization) and adds belief-update probes via perspective taking and false-belief variants. To disentangle exploration from reasoning, they compare active on-policy agents with passive runs over standardized proxy trajectories (SCOUT for vision, STRATEGIST with AC-3 for text) and score exploration efficiency via information gain or coverage. Results across several state-of-the-art models show strong text-world performance but large drops in vision, common failures in ego-motion updates and global consistency, and a tendency to terminate exploration too early, highlighting the need for agents that manage spatial belief as a first-class capability.

**Strengths:**

S1.  While the setup is admittedly a bit toyish, the core capability it targets—actively constructing, updating, and using a spatial belief from partial egocentric views—is exactly what real robots need to operate safely and robustly in cluttered, changing environments; from home assistants that must map and remap rooms, to warehouse and hospital robots that navigate multi-room spaces with moving obstacles, the paper’s emphasis on belief management, uncertainty reduction, and allocentric–egocentric conversion translates directly into high-value, real-world behaviors.

S2. The paper cleanly isolates “active belief construction under uncertainty” as the missing capability between passive reasoning and task-driven foraging, giving the field a crisp target. Also, they require stepwise cognitive-map JSON forces models to externalize internal spatial state, letting evaluators inspect correctness, consistency, and stability—not just final-task accuracy.

S3. Dual-modality design: mirrored text and vision worlds over identical layouts quantify the real modality gap and let researchers attribute errors to perception vs. spatial inference.

**Weaknesses:**

W1. Discrete bins (5 egocentric angle bins, 6 distance bins, 8 allocentric directions) compress geometry in ways that can hide small-but-critical errors (or inflate accuracy by near-miss snapping); no sensitivity analysis shows how bin granularity changes rankings or conclusions.

W2. Layouts are grid-based with tree-structured room graphs (no loops), weakening tests of global consistency (e.g., cycle closure) and long-range metric drift—the very failure modes cited (egomotion, global map maintenance) would be more exposed in loopy topologies and irregular geometries.

W3. The authors grant an action that returns ground truth, acknowledge cost-minimization under a step budget, and evaluate information gain via AC-3 in text mode—yet they don’t ablate Query usage, price, or availability. That omission makes it hard to know how much of the reported “spatial belief construction” truly reflects inference from partial observations versus selective peeks at the answer.

**Questions:**

The statistical robustness is unclear: the paper mentions 100 seeds but not per-task item counts, variance across seeds, or multiple-comparison controls. Given many tasks and modalities, some reported gaps could reflect sampling and prompt stochasticity rather than stable capability differences.

---

> ### Author Response · Authors · 2025-11-22
> **Response to Reviewer TnFX (1/2)**
>
> We thank the reviewer for the encouraging assessment and for recognizing that ToS addresses a critical, high-value real-world capability. We appreciate the constructive feedback on our experimental design and address these points below to further demonstrate the robustness of our findings.
>
> ---
>
> ## Response to Weakness
>
> **1. [W1] Effect of bin granularity**
>
> We thank the reviewer for the comments on bin granularity and potential sensitivity. To address this, we reran experiments on 20 samples with finer bins: 15° direction bins (vs. 22.5°) and distance bins of 1.5, 3, 6, 12, and 24 units. With this finer scheme, GPT-5 (0.82 vs. 0.84) and GEMINI-2.5-PRO (0.77 vs. 0.75) show similar performance to the paper, while CLAUDE-4-SONNET improves (0.46 vs. 0.41). The overall model ranking is unchanged, indicating bin granularity has only a minor effect.
>
> **Table 1. Results across different models using bin system with a different granulrity**
>
> | Model | Exp.cost |  Avg. |direction | act2view | view2act | alloc.map | ment.rot | loc2view | rot.dir | view2loc |
> |--------|----------|-----------|----------|----------|-----------|----------|----------|---------|----------|------|
> | GPT-5 | 6.85     | **0.815** | 0.683     | 0.783    | 0.617    | 0.967     | 0.833    | 0.883    | 0.933    | 0.817    |
> | GEMINI-2.5-PRO |10.95     | **0.769** | 0.717     | 0.667    | 0.800     | 0.750     | 0.767    | 0.733     | 0.967    | 0.750     |
> | CLAUDE-4-SONNET | 10.3     |  **0.463** | 0.317     | 0.450    | 0.383    | 0.617     | 0.383    | 0.683    | 0.600    | 0.267     |
>
> **2. [W2] Effect of Loop topology**
>
> We appreciate the reviewer’s suggestion regarding topological complexity. To investigate whether tree structures trivialize the task compared to loopy topologies, we conducted an ablation study using GPT-5 (20 samples). We compared a 4-room Tree (Center connected to 3 leaves) against a 4-room Loop (Cyclic: 1↔2↔3↔4↔1).
>
> Contrary to the intuition that loops are more difficult due to cycle closure, our results indicate that the Loop topology actually yields higher performance and lower exploration costs compared to the Tree. We attribute this to two factors:
>
> + **(2.1). Exploration Efficiency.** In a loop, the agent can visit new areas continuously without backtracking. In the Tree structure, the agent is forced to "backtrack" (traverse A→B→A) to explore different branches. This backtracking forces the agent to manage egomotion errors over longer trajectories to return to the decision point, thereby rigorously testing metric consistency.
> + **(2.2). Implicit Loops in Trees.** Exploring a tree naturally creates “traversal loops” (going to a leaf and back). The agent must recognize the central room on return to continue to the next branch, effectively testing global consistency and map maintenance much like a geometric loop.
>
> **Table 2. Results of 4-room with different topologies**
>
> | Topology| Exp. cost | Avg.   | direction | act2view | view2act | alloc.map | ment.rot | loc2view | rot.dir | view2loc |
> |----------------|-----------|----------:|---------:|---------:|----------:|---------:|---------:|--------:|---------:|-------:|
> | Tree     | 21.5    | **0.529** | 0.500     | 0.583    | 0.450     | 0.267     | 0.667    | 0.567     | 0.700    | 0.500     |
> | Loop  | 19.2     | **0.548** | 0.500     | 0.483    | 0.283    | 0.283     | 0.800    | 0.733    | 0.800    | 0.500    |
>
>
> **3. [W3] Ablation of Query action**
> We thank the reviewer for the insightful comment. We realize that we did not state very clearly in the paper that the Query action has a cost of 2 and the Observe action has a cost of 1. We will make this explicit in the revised version.
>
> +  In our experiments we observed that when the query cost is 2, the model almost never invokes the action (less than $1\%$), which means that the model truly construct their internal spatial belief from partial observations instead of peeks at the answer.
> + We also varied the query cost (0.5 and 1) to assess its impact on active exploration. With a cost of 1, the model continues to avoid the query action, matching the paper’s reported performance. Reducing the cost to 0.5 leads the model to query about 0.4 times per sample, only for previously unseen objects. The resulting Table 3 scores are slightly higher than in the original paper, but not statistically significant.
>
>
> **Table 3. Results of Query cost = 0.5 (original avg. acc is 0.83 for the same subset)**
>
> | Exp. cost |  Avg.   | direction | act2view | view2act | alloc.map | ment.rot | loc2view | rot.dir | view2loc |
> |-----------|----------:|---------:|---------:|----------:|---------:|---------:|--------:|---------:|-------:|
> | 7.3       | **0.856** |    0.867  |   0.883  |   0.633  |    0.883  |   0.933  |   0.900  |   0.900 |   0.850  |

---

> ### Author Response · Authors · 2025-11-22
> **Response to Reviewer TnFX (2/2)**
>
> ## Response to Questions
>
> **1. [Q1] statistics**
>
> We appreciate the reviewer's focus on statistical rigor. To clarify our protocol:
> 1. For each model and each task, we evaluate performance on 100 distinct environments (seeds 0–99). In every environment, we generate three unique queries, yielding $N = 300$ evaluation instances per task per model, or $N = 2400$ total questions for each model across all eight tasks.
> 2. Paired Design via Seed Tying: As detailed in Appendix B, we employ "seed tying" to ensure bit-for-bit reproducibility. Every model faces the exact same set of 300 layouts and questions. This functions as a paired comparison design, isolating model capability as the sole variable and eliminating environmental variance between models.
> 3. We agree that clearer reporting of statistical reliability is important. In the camera-ready version, we will add full statistical analyses across all tasks

---

> > ### Author Response · Authors · 2025-11-26
> > **Rebuttal Follow up**
> >
> > Dear Reviewer TnFX,
> >
> > We have posted our response to your initial review to address the weaknesses and questions you rainsed. As the discussion period progresses, we would very greatful if you could take a moment to look over our rebuttal. We would be glad to respond to any further questions or give additional clarification as needed.
> >
> > Your insights are extremely valuable in helping us refine the final version of our work. Thank you once again for your time and thoughtful evaluation.
> >
> > Thanks, Authors

---

### Official Review · Reviewer_9VR4 · 2025-10-28

**Soundness:** 2
**Presentation:** 4
**Contribution:** 3
**Rating:** 6
**Confidence:** 4

**Summary:**

This paper introduces a new Theory of Space framework that tests the agent's ability to learn the unobserved structure of its spatial environment from partial observations.
They implement a benchmark including text-based and visual environments for curiosity-driven exploration, and prompt (SOTA foundation) models to present their cognitive map at each step to track the underlying spatial model.
They document common failure modes in how models' form spatial beliefs, including path integration errroe and incorrect map inference.
The primary focus of the benchmark is exploration: the agent must make its own decisions on what to observe next.
They find that the models perform much better in pure text-based worlds compared to visual ones.

**Strengths:**

Theoretical soundness: The paper is well written and well grounded in cognitive theory.

Technical soundness: A stack of tools and technologies is integrated effectively to design the ToS benchmark. A large number of models is evaluated

**Weaknesses:**

This work aims to produce a benchmark for human-like spatial understanding and exploration. While this motivation, and the ToS approach as a means of achieving it are very well theoretically justified, the paper offers no evidence of how a human would behave in these tasks.

The models are compared to two proxy agents, which they claim to execute a theoretically optimal path, to establish an upper bound on exploration ability.  However, no proof of proxy agent optimal is given.

Both weaknesses arise from overstated claims. While would take a lot of work to support these claims, the paper would be still valuable if the claims were more accurately reframed.

**Questions:**

How would foundation models compare to human performance?

Is it possible to design a human study where people navigate the same worlds?

It is worth considering that while human map drawings are notoriously poor, their navigation abilities are often excellent. Have the authors considered to what extent this pattern of knowledge accessibility may be parallelled in machines?

---

> ### Author Response · Authors · 2025-11-22
> **Response to Reviewer 9VR4**
>
> We thank the reviewer for the positive assessment of our paper’s presentation and theoretical grounding, and for the constructive feedback regarding our claims. We address weaknesses and questions below.
>
> ---
>
> ## Response to Weakness
>
> **1. [W1] human performance**
>
> We thank the reviewer for this insightful comment. To provide extra human baselines, we built a optimized web interface and ran a small-scale study.. Specifically, we recruited $3$ human annotators and asked each to complete $5$ text-world environments and $5$ vision-world environments.
>
> As shown in Table 1, humans achieve near-perfect performance across all subtasks, confirming that the proposed benchmark is solvable by human participants and that it reflects realistic spatial understanding and exploration capabilities. Importantly, the performance gap between humans and current SOTA foundation models remains substantial, particularly in vision-based environments ($0.97$ vs. $0.6$), highlighting the benchmark’s challenge and relevance.
>
> We also observe that humans incur a lower exploration cost in vision environments compared to text environments. This aligns with cognitive findings that visual information is more intuitive and less demanding to process than symbolic textual descriptions.
>
> **Table 1 Human performance for text and vision world**
> | World | Exp.cost | Avg   | direction | act2view | view2act | alloc.map | ment.rot | loc2view | rot.dir | view2loc |
> |----------|----------|--------|-----------|----------|-----------|-----------|----------|-----------|----------|-----------|
> | Vision   | 7.067    | **0.967** | 0.956     | 0.956    | 1.000     | 0.956     | 1.000    | 0.911    | 1.000    | 0.956     |
> | Text     | 8.933    | **0.978** | 1.000     | 1.000    | 0.933     | 1.000     | 1.000    | 0.889    | 1.000    | 1.000     |
>
>
> **2. [W2] Proxy Agent Optimality**
> We agree with the reviewer that the term "theoretically optimal" was an overstatement, as we did not provide a formal proof of global optimality. We will revise the descriptions to better reflect the underlying algorithms. specifically: The Strategist (Text World) follows a greedy approach to maximize information. It prioritizes objects with the highest uncertainty to reduce ambiguity fast. While effective, this heuristic doesn't guarantee the global shortest path. The Scout (Vision World) relies on exhaustive search (visit-sweep-advance). This ensures complete coverage, even if it isn't always the most step-efficient solution."
> However, We want to highlight that these proxies are meant to set an empirical upper bound for exploration success, rather than efficiency. These proxies successfully achieve $1.0$ (complete) information gain, whereas current state-of-the-art models plateau at $\approx 0.8$. Thus, they successfully fulfill their role in the benchmark: disentangling reasoning capabilities from exploration failures by providing fully explored trajectories.
>
> ---
>
> ## Response to Questions
>
> **1. [Q1] human performance**
>
> Same as **W1**
>
> **2. [Q2] platform for human study**
>
> Thank you for this question. Yes, it is not only possible, but we have already implemented a web-based evaluation platform and conducted a human study to establish a performance upper bound.
> To ensure fairness, humans used the exact same Gym interface as the models. They navigated the same worlds with the same view constraints (step-by-step movement) and answered the same questions. The only difference is that humans were tested on a subset of environments. We will add screenshots of the UI to the Appendix.
>
> **3. [Q3] navigation vs. map drawing**
>
> We really appreciate the cognitive science parallel. In fact, we explicitly designed our framework to capture this distinction, separating 'Belief on Route' (navigation) from 'Belief on Survey' (map drawing).
> Our results confirm that foundation models share this human-like discrepancy, especially during active exploration. By treating Allocentric Mapping as 'map drawing' and View2Action/Action2View as 'navigation,' we see a clear pattern:
> Top models (like GPT-5 and GEMINI-2.5-PRO) consistently outperform on navigation tasks compared to metric mapping (Table 1 in paper). For instance, GPT-5 scores $0.59$ on View2Action (navigation) versus only $0.48$ on map drawing. This suggests that, just like humans, models perform better on egocentric 'route knowledge' than on precise allocentric 'survey knowledge'.

---

> > ### Author Response · Authors · 2025-11-26
> > **Rebuttal Follow up**
> >
> > Dear Reviewer 9VR4,
> >
> > We have posted our response to your initial review to address the weaknesses and questions you rainsed. As the discussion period progresses, we would very greatful if you could take a moment to look over our rebuttal. We would be glad to respond to any further questions or give additional clarification as needed.
> >
> > Your insights are extremely valuable in helping us refine the final version of our work. Thank you once again for your time and thoughtful evaluation.
> >
> > Thanks, Authors

---

### Official Review · Reviewer_dNos · 2025-10-28

**Soundness:** 3
**Presentation:** 4
**Contribution:** 3
**Rating:** 8
**Confidence:** 3

**Summary:**

This paper introduces the Theory of Space (ToS), a framework for evaluating the ability of foundation models to actively construct, update, and utilize an internal spatial belief of an environment from partial observations. The paper designed experiments to (1) ask an LLM/VLM agent to actively explore a partially observable environment for establishing the internal spatial model, (2) use a proxy agent script to derive a ``passive" spatial model to disentangle active exploration from passive reasoning, and (3) ask the agent to represent its spatial belief for direct accessment. Evaluations across modern LLM/VLMs show a modality gap (stronger on text than vision) and diagnose failure modes like egomotion update errors and poor global map maintenance; humans outperform most models with modest exploration budgets.

**Strengths:**

1. The paper designs spatial belief evaluation around task-agnostic, uncertainty-reducing exploration, rather than passive reasoning and goal-directed task completion, adding an active aspect to spatial belief construction. This makes the paper original.
2. Spatial understanding under partial observability is central for embodied agents and planning. The ToS framework fills in the gap for LLM evaluation in enactive cognition by evaluating the goal-agnostic exploration of active LLM agents.
3. Each experiment in the benchmark is well-designed for evaluating the representation, updating, and utilization of the internal space model. The metrics used for measuring the performance are reasonable, including the exploration efficiency against the AC-3 algorithm, belief quality assessment by correctness and consistency, and task success rate.

**Weaknesses:**

1. Relatively simple spatial environment and limited statistical results: most experiments use two connected $6$ by $6$ rooms with $9$ objects, and some with varying room size. While the small size of the spatial environment avoids memory capacity as a confounding factor for ToS performance, its simplicity could potentially trivialize the active exploration aspect of the agent. The generalization beyond the grid room is therefore unclear. As a consequence, there are also no statistical results on the performance.
2. Exploration efficiency analysis: the exploration efficiency is mainly analyzed by the information-gain curve. It's unclear whether limited exploration thoroughness originates from policy or belief integration failures.

**Questions:**

1. How do you ensure the cognitive map probing reflects the true internal state rather than post-hoc rationalization?
2. Is the termination action fully decided by the LLM agent? What could be the cause that leads to agents stopping prematurely?

---

> ### Author Response · Authors · 2025-11-22
> **Response to Reviewer dNos (1/2)**
>
> We sincerely thank the Reviewer for recognizing the "originality" of our task-agnostic exploration framework. We appreciate the acknowledgment that our ToS framework "fills in the gap" for evaluating enactive cognition and that our experiments are "well-designed." We address weakness and questions below.
>
> ---
>
> ## Response to Weakness
>
> **1. [W1] Room setting too simple**
>
> We thank the reviewer for raising the important question of generalization. We address this in two parts: justifying the baseline complexity and providing new results on larger-scale environments.
>
> + **(1.1). Baseline Difficulty and Inclusivity.** We clarify that the 2-room setting remains challenging, particularly in the vision domain. As shown in Table 1 of the paper, even state-of-the-art models like GPT-5 and GEMINI-2.5-PRO achieve only $\approx 0.60$. Furthermore, we selected this scale to ensure a fair comparison across a spectrum of model capabilities. While strong models explore efficiently ($\approx 8$ steps), other models (e.g., CLAUDE-4-SONNET, GLM-4.5V) require higher costs ($>15$ steps). The 2-room setting is basically a reasonable baseline. It allows us to evaluate less capable models without having to deal with context or step limits.
> + **(1.2). Results for more complex room setting** In order to check capacity limits for powerful models like GPT-5, we add two more experiments with more challenging room settings: 3-room and 4-room. For our environment is simulator-based and gym-like, it's easy to scale it to different room settings. Additionally we design complicated topology for these two room settings, where the main room is connected to all other rooms, so after exploring one room, the agent needs to first return to the first room and goes to another. We use GPT-5 and 20 samples to conduct experiments. We show the 3-room results in Table 1 & 4-room results in Table 2. Here we include results of both active and passive for both text and vision world.
> + **(1.3). Analysis.** The results from the 3-room and 4-room experiments strongly reinforce our paper's central hypothesis: active exploration capability is a bottleneck.
>     - Widening Active-Passive Gap: There is a persistent performance gap between Active and Passive settings. Crucially, this gap widens as environmental complexity increases. In the Vision-based world, the gap expands from $\approx 7.9\%$ in the 3-room setting ($0.556$ vs. $0.635$) to $\approx 11.5%$ in the 4-room setting ($0.529$ vs. $0.644$).
>     - Scaling Difficulty: As the topology becomes more complex, the exploration cost rises significantly (e.g., from $13.8$ to $21.45$ in Vision), yet the active agent's ability to construct an accurate map degrades faster than its passive reasoning baseline. This confirms that while models like GPT-5 can reason effectively over "perfect" exploration logs, they struggle to autonomously gather the necessary information in complex spaces, validating the importance of the ToS framework.
>
> **Table 1. Performance of 3-room settings**
>
>
> | Exp setting  | Modality | Exp. cost | Avg.   | direction | act2view | view2act | alloc.map | ment.rot | loc2view | rot.dir | view2loc |
> |----------------|----------|-----------|----------:|---------:|---------:|----------:|---------:|---------:|--------:|---------:|-------:|
> | Active| Text     | 11.3     | **0.815** | 0.767     | 0.733    | 0.633     | 0.883     | 0.800    | 0.900     | 0.933    | 0.867     |
> | Passive  | Text     | -         |  **0.860** |0.700     | 0.850    | 0.850     | 0.883     | 0.900    | 0.767     | 1.000    | 0.933     |
> | Active | Vision     | 13.8     | **0.556** | 0.433     | 0.600    | 0.400    | 0.383     | 0.633    | 0.617    | 0.900    | 0.483     |
> | Passive  | Vision     | -         | **0.635** |     0.400     | 0.650    | 0.650     | 0.417     | 0.767    | 0.650     | 0.933    | 0.617     |
>
> **Table 2. Performance of 4-room settings**
>
> | Exp setting        | Modality | Exp. cost | Avg.   | direction | act2view | view2act | alloc.map | ment.rot | loc2view | rot.dir | view2loc |
> |----------------|----------|-----------|----------:|---------:|---------:|----------:|---------:|---------:|--------:|---------:|-------:|
> | Active| Text     | 16.55    | **0.773** | 0.667     | 0.733    | 0.633     | 0.650     | 0.833    | 0.933     | 0.917    | 0.817     |
> | Passive | Text     | -         | **0.827** |0.717     | 0.850    | 0.583     | 0.850     | 0.933    | 0.817     | 1.000    | 0.867     |
> | Active| Vision     | 21.45    | **0.529** | 0.500     | 0.583    | 0.450     | 0.267     | 0.667    | 0.567     | 0.700    | 0.500     |
> | Passive | Vision  | -         | **0.644** | 0.600     | 0.583    | 0.617     | 0.233     | 0.850    | 0.767     | 0.950    | 0.550     |

---

> ### Author Response · Authors · 2025-11-22
> **Response to Reviewer dNos (2/2)**
>
> **2. [W2] Source of exploration inefficiency unclear**
>
> We thank the reviewer for the chance to clarify this. We agree that distinguishing between policy inefficiency and belief integration failure is critical.
>
> While the information-gain curve gives a high-level view, we actually disentangle these factors through two specific mechanisms (Sec 2.1 & 3.4), which we will clarify in the revision:
>  + Decoupling via Proxy Agents: We use proxy agents to generate exploration logs.
>     + Passive Setting: Since the proxy guarantees information coverage, any failure here is strictly a reasoning failure (Belief Integration), not an exploration one.
>     + Active Setting: By comparing this against the Active setting (where models plan their own paths), we isolate the specific performance gap caused by exploration limitations. This gap is clearly shown in the 3-room and 4-room results (Tables 1 & 2).
> + Direct Analysis: Beyond the curve, Section 5.2 provides a separate analysis that directly probes the belief-formation process."
>
> ---
>
> ## Response to Questions
>
> **1. [Q1] Does probing reflect true state**
>
> Regarding post-hoc rationalization, we want to clarify that our design treats the cognitive map as a real-time snapshot of the agent's state, not as an explanation generated after the fact
>
> + **(1.1). State-Based Probing, Not Justification.** We do not ask the model to explain why it took an action (which often leads to rationalization). Instead, we prompt the model to strictly translate its current observation history into a cognitive map. This ensures the output reflects the information currently available in the model's context window.
> + **(1.2) Independence of Generation.** As noted in Section 5.1, the cognitive map is probed independently at each step. During this probing phase, the model is provided only with its exploration history and can not see previous cognitive map or evaluation tasks. This prevents the model from copying previous hallucinations or conditioning its map on desired downstream outcomes.
>
> **2. [Q2] Question about termination**
>
> + **(2.1). Autonomy.** Yes, the termination action is fully autonomous. The agent must explicitly decide to stop exploration based on its own internal assessment of the environment. In our experiments, models rarely exhausted the step budget; for instance, GPT-5 and GEMINI-2.5 PRO averaged fewer than (or equal to) $10$ steps, while the maximum exploration budget is $20$ steps.
> + **(2.2).Premature Termination.** We attribute premature stopping to a failure in monitoring uncertainty. In the Text World, observations are categorical and inherently ambiguous (e.g., "mid-distance" covers a range of cells). Resolving this requires multi-view observation of the same object to eliminate potential object positions. As shown in the accumulated info-gain curves, models often plateau at $\approx 0.8$, whereas the "Strategist" proxy achieves $1.0$. The problem is that the model thinks 'I've seen it' means 'I know where it is.' It terminates early because it confuses visibility with certainty, failing to realize the location is still ambiguous.

---

> > ### Author Response · Authors · 2025-11-26
> > **Rebuttal Follow-up**
> >
> > Dear Reviewer dNos,
> >
> > We have posted our response to your initial review to address the weaknesses and questions you rainsed. As the discussion period progresses, we would very greatful if you could take a moment to look over our rebuttal. We would be glad to respond to any further questions or give additional clarification as needed.
> >
> > Your insights are extremely valuable in helping us refine the final version of our work. Thank you once again for your time and thoughtful evaluation.
> >
> > Thanks,
> > Authors

---

### Official Review · Reviewer_uS5h · 2025-10-31

**Soundness:** 2
**Presentation:** 3
**Contribution:** 2
**Rating:** 4
**Confidence:** 4

**Summary:**

The paper introduces Theory of Space (ToS), an evaluation paradigm for whether LLM/VLM agents can actively construct, update, and use an internal spatial belief (a cognitive map) through exploration. It builds a procedural multi-room world (text and vision), defines route/survey/update tasks, decouples exploration from reasoning using proxy agents, and probes the belief itself (serialized maps) rather than only task success. Empirically, top models perform well in text but struggle in vision. Active and passive performance are close, suggesting belief construction and maintenance (not coverage alone) is the bottleneck.

**Strengths:**

- Concepts and definition contribution: ToS captures the ability to (1) construct a globally consistent belief from partial views, (2) update it as new evidence conflicts, (3) utilize it for downstream spatial tasks. The two-phase evaluation separates Exploration to gather observations and build a belief from Reasoning on route/survey/update tasks to test utilization. The task formulation itself is interesting, although this paradigm has been explored in early embodied AI benchmarks like EXCALIBUR (I knew EXCALIBUR evaluates where the agent goes; ToS evaluates what the agent infers and remembers, there is a difference).
- Belief probing & metrics: Models output a cognitive map (JSON) that can be directly examined. The paper provides metrics for positional, directional, and facing accuracy, plus a composite score. It also introduces a formal Exploration Efficiency metric based on information gain with AC-3 pruning.
- Interesting findings: Active performance roughly equals passive performance (but with premature stops). Active scores are close to passive, with gaps often stemming from premature termination and inefficient exploration, not pure lack of path coverage (see information-gain curves). Belief management matters: stepwise probing shows errors compound from perception to integration (egomotion) to stability. Text models rank higher on correctness/consistency, while in vision, perception is the bottleneck.

**Weaknesses:**

- The current environment relies on symbolic discretization: angles and distances are bucketed into categorical bins (e.g., {near, mid, far}) and rendered with calibration cues (reference grids, constant lighting). This yields clean supervision but strips away realistic sensory ambiguity such as partial occlusions, depth uncertainty, and texture variation. Models may appear spatially consistent only because the environment removes real-world ambiguities. This inflates performance and may not transfer to continuous or noisy perception (e.g., real egocentric videos or robot camera feeds).
    - Possible action items: Add a continuous-valued variant where direction and distance are real-valued regressions rather than categorical bins; introduce sensor noise and texture perturbations (random lighting, viewpoint jitter, partial occlusion).

- The results show a significant gap between text and vision performance, but the source of this gap (perceptual encoding vs. multimodal integration vs. belief updating) remains ambiguous. Without identifying the bottleneck, it's unclear whether improvements should target visual representation learning, grounding alignment, or belief maintenance logic.
    - Possible action items: Implement modular ablations: Perfect-perception oracle (feed ground-truth object positions); Perfect-integration oracle (feed clean percepts, test memory/belief fusion); Perfect-stability oracle (test only inference drift).

- Despite the paper's emphasis on ToS, the core scientific question (can an embodied agent build a consistent internal map from partial observations) closely parallels prior work such as EXCALIBUR (CVPR 2023) and Emergence of Maps in the Memories of Blind Navigation Agents (NeurIPS 2023). Both earlier studies explored spatial map emergence and information-gain-based exploration. The conceptual novelty is less about introducing a new phenomenon and more about repurposing it into an evaluation benchmark. Without explicit differentiation, readers may view ToS as incremental rather than foundational.

**Questions:**

- Space and ToM has been extensively studied in past psychology research. See https://arxiv.org/abs/2310.19619 for example. Your benchmark includes "false-belief / perspective-taking" tasks, but these seem to involve the agent's own belief about the environment rather than what another agent believes, i.e., ToS did address spatial belief of a single agent about objects and its own perspective, but did not appear to include multiple agents with independent goals or belief states. How do you justify this choice?
- Can ToS-style belief actually improve downstream navigation/manipulation? JSON is a strong choice, and vision seems to play a minimal role here.
- How sensitive are results to Query and termination policy? Could Query or early-stop heuristics be creating artificial gaps between active and passive? Maybe provide cost-sweeps over Query availability/penalties and termination criteria; plot accuracy vs. action cost curve?

 - - -
I am happy to change my ratings if the above Weaknesses and Questions are addressed.

---

> ### Author Response · Authors · 2025-11-22
> **Response to Reviewer us5h (1/4)**
>
> We sincerely thank the Reviewer for their detailed feedback and for recognizing our work's "concepts and definition contribution" and the value of our "belief probing & metrics." We appreciate the acknowledgment that our findings regarding active versus passive exploration and belief management are interesting.
> We address weakness and questions below.
>
> ---
>
> ## Response to Weakness
> 1. **[W1] Environmental Discretization and Realism**
>
> We appreciate the reviewer’s feedback and clarify that our design choices are deliberate: clean visual cues are used to disentangle spatial reasoning from perceptual noise, while symbolic text discretization is designed to induce uncertainty and align with human spatial cognition. Our new experiments with continuous values and visual perturbations confirm that our original setup accurately targets the core Theory of Space capability. By default, we use GPT-5 and select 20 samples from our dataset to conduct the experiments.
>
> + **(1.1). Continuous-valued relation expressions.** We employ categorical bins for two primary reasons. First, this design aligns with human cognitive processes, which rely on qualitative, topological relationships (e.g., 'near', 'left of') rather than precise metric coordinates. Second, discretization explicitly introduces state uncertainty: because a single bin corresponds to a range of possible locations, the model is forced to actively integrate multiple observations to resolve ambiguity, rather than relying on perfect metric perception. We add an experiment in which spatial relations are expressed using continuous textual values, for example: *"backpack: +14° from front, 4.12 away, facing left."* The active exploration results are shown in Table 1. This representation outperforms the categorical-bin version (average accuracy 0.84 in the paper; 0.83 on the same subset used here). The improvement is expected, as continuous-valued relations remove the ambiguity inherent in discretized bins, allowing the model to reason more precisely.
> + **(1.2). Purposefully noise-free vision and a jittered-vision ablation.** Our primary goal is to evaluate the model’s theory of space, that is, its ability to construct and use an internal spatial belief. To isolate this capability, we deliberately configure the vision environment to minimize perceptual noise, enabling us to disentangle spatial reasoning from perception errors.
> For completeness, we also include an experiment with different lights and background texture in Table 2. In our simulated environments, it's easy to change light, background, viewpoint settings.  In this experiment, we keep all scene factors fixed (room layout, object placement, doors, agent poses) and change only the lighting: we add a dusk-style skybox and lower the light height, producing more irregular illumination with stronger grazing-angle shadows and uneven brightness. This alters the visual appearance while preserving geometry, and the performance remains similar.  Results show that for the same scene but with a different visual appearance, performance for all three models improves a little, the trend and ranking are still consistent.
>
>
> **Table 1. results for continuous-valued textual observation (original avg. acc for same subset is 0.83)**
>
> | Exp.cost | Avg. | direction | act2view | view2act | alloc.map | ment.rot | loc2view | rot.dir | view2loc |
> |----------|-----------|----------|----------|-----------|----------|----------|---------|----------|------|
> | 6.450    | **0.940** | 0.950     | 0.917     | 0.767     | 1.000      | 0.967     | 1.000     | 0.950    | 0.967     |
>
>
> **Table 2. results for different light environments (original avg. acc for same subset for GPT-5, GEMINI-2.5-PRO, and CLAUDE-4-SONNET is 0.63, 0.61, 0.36)**
>
> | Model | Exp.cost | Avg. | direction | act2view | view2act | alloc.map | ment.rot | loc2view | rot.dir | view2loc |
> |-------|----------|-----------|----------|----------|-----------|----------|----------|---------|----------|------|
> | GPT-5 | 8.9      |**0.644** | 0.500     | 0.617    | 0.617    | 0.467     | 0.767    | 0.717    | 0.883   | 0.583    |
> | GEMINI-2.5-PRO | 10.85    | **0.629** | 0.367     | 0.633    | 0.683    | 0.500     | 0.667    | 0.667    | 0.967   | 0.550    |
> | CLAUDE-4-SONNET | 13.9     | **0.404** | 0.367     | 0.333    | 0.417     | 0.383     | 0.400    | 0.467     | 0.583    | 0.283     |

---

> ### Author Response · Authors · 2025-11-22
> **Response to Reviewer us5h (2/4)**
>
> 2. **[W2] Ambiguity of the Modality Gap**
>
> We thank the reviewer for this constructive suggestion to disentangle the sources of error. Following the recommendation, we implemented modular ablation studies with oracle baselines. Our results quantitatively confirm that perception is the primary bottleneck: when perceptual uncertainty is removed, the visual agent's performance substantially improves and aligns with the text-based setting.  We use GPT-5 and select 20 samples from our dataset to conduct the experiments.
>
> + **(2.1). Perfect-perception oracle (feeding ground-truth object positions and orientation).** We add an experiment in which the agent receives the ground-truth local cogmap during active exploration, including exact object positions and orientations, along with the vision inputs during exploration. The results, as shown in Table 3 (entry "gt-local"), reach 0.894, substantially higher than the average 0.59 reported in the paper. This large performance gap indicates that the primary bottleneck is perceptual quality: once the vision module is given perfect perception, the model’s performance becomes comparable to that of the text-observation setting (where perception is effectively perfect).
> + **(2.2). Perfect perception, integration, and stability ("oracle") experiment.** We conducted an experiment to assess the model’s ability when endowed with perfect perception, integration and stability, i.e., when it can build a flawless internal spatial belief. Specifically, we provided the model with a ground-truth global cognitive map and measured its performance under this "oracle" condition. As shown in Table 3 (entry "gt-global"), the model achieved a nearly perfect result of 0.975. This large improvement is expected, since with oracle spatial belief, the model can correctly answer almost all spatial questions.
>
>
> **Table 3. Perfect Perception and Perfect perception, integration, and stability.**
> | Methods | Exp.cost | Avg. | direction | act2view | view2act | alloc.map | ment.rot | loc2view | rot.dir | view2loc |
> | :--- | :---: | :---: | :---: | :---: | :---: | :---: | :---: | :---: | :---: | :---: |
> | gt-local | 7.750 | **0.894** | 0.950 | 1 | 0.700 | 0.950 | 0.850 | 0.900 | 0.850 | 0.950 |
> | gt-global | - | **0.975** | 1 | 0.900 | 0.950 | 1 | 1 | 1 | 0.950 | 1 |
>
> 3. **[W3] Parallel with existing work**
> We thank the reviewer for highlighting the parallels with existing work. We agree that inferring global structure from partial observations is a shared scientific foundation. However, we would like to highlight the differences. While prior works demonstrated that specialized RL agents can learn to navigate or answer questions, ToS asks a fundamentally different question: Do general-purpose foundation models possess an explicit Theory of Space?
>
> ToS differs from these works in three critical dimensions:
>
> + **(3.1). Foundation Model Reasoning vs. RL Policy Learning.** Prior work such as EXCALIBUR trains embodied agents with RL over millions of steps to learn task-specific navigation policies. In contrast, ToS evaluates pre-trained Foundation Models (LLMs/VLMs) in a zero-shot setting to diagnose their inherent spatial reasoning abilities. Our goal is not to assess learnable policies, but to test whether these models can construct a world model from context without task-specific fine-tuning. A key difference lies in the optimization objective: RL agents receive reward signals that effectively reveal the evaluation criteria and make exploration goal-driven. ToS instead evaluates models in a strictly task-agnostic regime, requiring them to form general-purpose spatial beliefs without prior knowledge of the downstream tasks.
> + **(3.2). Explicit Belief Probing vs. Task Performance.** EXCALIBUR evaluates spatial understanding implicitly through downstream VQA accuracy. As noted in prior literature, high VQA scores can sometimes result from dataset biases or shortcuts rather than true spatial understanding. Emergence of Maps analyzes implicit hidden states (a post-hoc analysis because of training). ToS introduces Explicit Probing of the internal belief. We force the agent to externalize its cognitive map at every step. This allows us to disentangle reasoning failures from exploration failures and directly measure the fidelity of the agent's internal representation, rather than just its behavioral success.
> + **(3.3). A Complete Cognitive Pipeline: Construction, Updating, and Utilization.** We formalize a task-agnostic pipeline that goes beyond exploration. Unlike previous benchmarks that focus largely on static map construction or navigation success, ToS specifically tests Belief Updating, which is the ability to revise the internal map when the environment changes ("false-belief"). This mirrors the "Theory of Mind" development in cognitive science  and provides a rigorous test of whether the agent maintains a robust, dynamic world model, a dimension largely absent in standard navigation benchmarks.

---

> ### Author Response · Authors · 2025-11-22
> **Response to Reviewer us5h (3/4)**
>
> ## Response to Questions
> **1. [Q1] Concern regarding the absence of a multi-agent setting.**
>
> We thank the reviewer for this insightful observation. We wish to clarify that our "Theory of Space" (ToS) framework is proposed as a cognitive counterpart to Theory of Mind (ToM), rather than an extension of it into the social domain.
> As detailed in Section 2, our analogy rests on the management of hidden states under uncertainty. In ToM, the hidden state is another agent's unobservable mind; in ToS, the hidden state is the unobservable global structure of the environment (which cannot be seen fully from any single viewpoint).
> We adapt ToM concepts to a spatial domain (belief vs. reality) rather than a social one (belief vs. belief):
> - False Belief: As detailed in section 5.3, this tests the agent's ability to update its internal map when it conflicts with current observations (e.g., a secretly moved object), mirroring the cognitive structure of the Sally-Anne test.
> - Perspective Taking: We define this as simulating observations from a hypothetical pose ("standing in another place") analogous to the social "standing in another's shoes." This usage is standard in recent spatial reasoning literature.
>
> **2. [Q2] ToS-style belief for downstream tasks.**
>
> We appreciate the opportunity to clarify the role of the cognitive map and the visual modality in our framework.
>
> + **(2.1). Map as a Diagnostic Probe, not a Prompting Strategy.** The primary function of the JSON cognitive map in ToS is to explicitly probe the agent's internal spatial belief state, rather than to serve as a Chain-of-Thought (CoT) intermediate step to boost performance. We aim to measure what the agent knows about the space, distinct from its ability to perform a specific task.
> + **(2.2). Additional experiments** We additionally include an experiment where the prompt is modified to require the model to first output a global cognitive map before giving the final answer (Table 4). We use GPT-5 and conduct experiments on a subset of 20 samples. The results remain nearly identical to those obtained under the original settings, indicating that explicitly generating a cognitive map does not yield measurable improvement. A plausible explanation is that GPT-5 already performs similar internal spatial reasoning steps within its hidden reasoning tokens, making the externalized cognitive-map step redundant.
> + **(2.3). The Role of Vision.** Vision plays a central and challenging role. Our benchmark uses a parallel text-vision environment to diagnose failure modes. As shown in Table 2 in the paper, there is a significant performance gap between text and vision (0.92 vs. 0.60 for GPT-5). The JSON map allows us to pinpoint the failure reason. Thus, the JSON format is essential for isolating where vision-based agents fail.
>
> **Table 4. Results of using Cognitive map as reasoning before final (original avg. acc = 0.59)**
>
> | Avg. | direction | act2view | view2act | alloc.map | ment.rot | loc2view | rot.dir | view2loc |
> | :---: | :---: | :---: | :---: | :---: | :---: | :---: | :---: | :---: |
> | **0.59** |  0.550 | 0.500 | 0.500 | 0.500 | 0.550 | 0.550 | 0.800 | 0.750 | 0.59|

---

> ### Author Response · Authors · 2025-11-22
> **Response to Reviewer us5h (4/4)**
>
> **3. [Q3] Effect of Query and termination policy**
>
> We thank the reviewer for the insightful comment. We realize that we did not state very clearly in the paper that the Query action has a cost of 2 and the Observe action has a cost of 1. We will make this explicit in the revised version. For the following experiments, we use GPT-5 and conduct on a subset of 20 samples.
>
> + **(3.1). Query and early-stop heuristics are not gap between active and passive.** First, in our experiments we observed that when the query cost is set to 2, the model almost never invokes the action. In addition, the average number of exploration steps is 6.7 for active exploration in the textual world and 8 for vision world, which is far below the maximum allowed exploration steps. This indicates that, compared to passive exploration, the model already has sufficient exploration steps available in the active setting.
> + **(3.2). Additional results for different query costs.** Additionally, we varied the query cost (0.5 and 1) to examine its effect on active exploration. When the query cost is 1, the model still refrains from calling the action, resulting in performance identical to that reported in the paper. When the query cost is reduced to 0.5, the model calls the query action an average of 0.4 times per sample, and only in cases where it has not previously encountered certain objects. The final results in Table 5 are slightly better than those reported in the original paper, but the improvement is not statistically significant.
> + **(3.3). Additional results for different exploration steps.** In the paper, we set the maximum number of exploration steps to 20. However, we observed that GPT-5 typically uses far fewer steps in practice: an average of 6.7 steps in the text setting and 8 steps in the vision setting. Based on this observation, we reduced the maximum exploration budget to 3 and 5 steps, respectively, and report the corresponding results in the table. These results show that increasing the exploration budget improves the model’s spatial understanding.
>
>
> **Table 5. Results of Query cost = 0.5 (original avg. acc is 0.83 for the same subset)**
>
> | Exp. cost |  Avg.   | direction | act2view | view2act | alloc.map | ment.rot | loc2view | rot.dir | view2loc |
> |-----------|----------:|---------:|---------:|----------:|---------:|---------:|--------:|---------:|-------:|
> | 7.3       | **0.856** |    0.867  |   0.883  |   0.633  |    0.883  |   0.933  |   0.900  |   0.900 |   0.850  |
>
> **Table 6. Effect of different max exploration steps (original avg. acc for the same subset: 0.83 for text, 0.63 for vision)**
>
> | Max steps        | Modality | Exp. cost | Avg.   | direction | act2view | view2act | alloc.map | ment.rot | loc2view | rot.dir | view2loc |
> |----------------|----------|-----------|----------:|---------:|---------:|----------:|---------:|---------:|--------:|---------:|-------:|
> | 3   | Text     | 3         | **0.656** |    0.417  |   0.683  |   0.517  |    0.867  |   0.667  |   0.583  |   0.850 |   0.667  |
> | 3   | Vision   | 3         | **0.537** |    0.400  |   0.517  |   0.433  |    0.417  |   0.600  |   0.617  |   0.850 |   0.467  |
> | 5   | Text     | 4.9       | **0.765** |    0.617  |   0.800  |   0.633  |    0.817  |   0.800  |   0.800  |   0.917 |   0.733  |
> | 5   | Vision   | 4.75      | **0.596** |    0.500  |   0.533  |   0.533  |    0.567  |   0.667  |   0.667  |   0.833 |   0.467  |

---

> > ### Author Response · Authors · 2025-11-26
> > **Rebuttal Follow up**
> >
> > Dear Reviewer uS5h,
> >
> > We have posted our response to your initial review to address the weaknesses and questions you rainsed. As the discussion period progresses, we would very greatful if you could take a moment to look over our rebuttal. We would be glad to respond to any further questions or give additional clarification as needed.
> >
> > Your insights are extremely valuable in helping us refine the final version of our work. Thank you once again for your time and thoughtful evaluation.
> >
> > Thanks,
> > Authors

---

> > > ### Comment · Reviewer_uS5h · 2025-11-26
> > >
> > > Thank you for the update!
> > >
> > > I am raising my ratings to 6, but I am not fully convinced by W3 and Q1. I think those are still valid limitations that this work should acknowledge, but I also think this work is already self-contained itself as a paper.

---

> > > > ### Author Response · Authors · 2025-11-30
> > > > **Response to W3 and Q1**
> > > >
> > > > Thank you for increasing your rating. We would like to further clarify our position for W3 and Q1:
> > > >
> > > > **For W3**
> > > >
> > > > 1. EXCALIBUR’s exploration metric highlights "coverage", which means visiting as many states as possible. While TOS focuses on minimal cost active exploration. Our agent must form a hypothesis about the unseen space and only explore what is necessary to resolve uncertainty.
> > > > 2. For both EXCALIBUR and Emergence of Map , the "map" is implicitly baked into the neural weights through millions of trials with RL. The agent learns where to go to maximize reward, often benefiting from "goal leakage" (knowing the target implicitly helps shape the policy). While in our approach, we use Foundation Models as zero-shot planners. Our agents have no prior training on the specific environment and no implicit knowledge of the final goal.
> > > > 3. Previous works largely evaluate spatial understanding through downstream performance (e.g., VQA accuracy). We go a step further. We don't just evaluate the utilization of the belief (the final answer), we evaluate the construction process. By tracking the agent's internal belief state over time, we measure how the agent builds its mental model.
> > > >
> > > > **For Q1**
> > > >
> > > > We view ToS as a cognitive parallel to ToM. Both require modeling a 'hidden variable' distinct from what is currently seen:
> > > >  - In ToM, that variable is another agent's mind.
> > > >  - In ToS, it is the unobserved environment (e.g., the layout behind a wall).
> > > >
> > > > We intentionally excluded social dynamics to isolate this spatial component. We agree that the multi-agent scenario you described is better termed 'Theory of Mind in Space.' While that is a fascinating direction, it is distinct from our focus on modeling the space itself.

---

### Author Response · Authors · 2025-12-01
**General Response**

We sincerely thank the AC and reviewers for their time. We now responded to all comments. We are encouraged that Reviewer `uS5h` raised their score from 4 to 6 on Nov 26, a day before the leakage notifcation, contributing to our final score of 8 6 6 6. However, the other reviewers have not yet had the chance to engage in the discussion. Below, we summarize the major concerns we have addressed.


**1. Human performance** (`9VR4`)

We also ran a small user study with three people to check the benchmark. Using the same room layouts in both text and vision settings (see Table 1 for reviewer `9VR4`), humans got almost perfect scores, showing the task is realistic and solvable, and that there’s still a huge gap between humans and SOTA models—especially for vision.

**2. Ablation of Query Action** (`TnFX`, `us5h`)

Regarding the action costs, we clarify that Query is set to $2$ and Observe to $1$. Under this setting, we found the model rarely queries. This suggests it’s really building a cognitive map from observations rather than taking shortcuts.

We also checked different costs: at $1$ the behavior is basically the same, and even at $0.5$ the model only queries a bit more (about $0.4$ times per sample, mostly for new objects), which gives slightly higher scores but nothing statistically significant.

**3. Ablation of Room settings** (`dNos`)

**Concern**: the 2-room setting may be too trivial

**Response**:
- The 2-room setting in paper remains a significant challenge; even SOTA models (GPT-5, Gemini-2.5) only achieve $\approx 0.60$. It effectively benchmarks reasoning without hitting the context limits of smaller models.
- To push SOTA limits, we leveraged our scalable gym to add 3-room and 4-room experiments with star-shaped topologies that require complex backtracking.
- Active Exploration is the Bottleneck: As complexity grows, the performance gap between Active and Passive settings widens significantly. While models can reason over "perfect" logs, they struggle to explore and gather that information in complex spaces.

**4. Novelty and distinctions from prior work (e.g., EXCALIBUR, Emergence of Map)** (`us5h`)

We distinguish ToS from prior work in three key ways.
- Unlike EXCALIBUR’s focus on maximizing coverage, ToS prioritizes efficiency. Our agent doesn't just visit states; it hypothesizes and explores only what is necessary to resolve uncertainty.
- While RL-based methods bake maps into weights (often suffering from "goal leakage"), our approach uses Foundation Models as zero-shot planners with no prior training or implicit map knowledge.
- We go beyond downstream results (like VQA). We explicitly evaluate the construction process, tracking how the agent’s internal belief state evolves over time.

---

### Meta-Review · Area_Chair_dx7p · 2026-01-04

**Summary:**

This work studies the emergence of spatial beliefs in foundation models and describes a "theory of space", probed by a new benchmark. The reviewers generally appreciated a well motivated paper; the idea to explicitly probe the models understanding of space; well-designed experiments; interesting findings; a well written paper.

Weaknesses raised were:
- Simplified environment involving symbolic discretization
- missing analysis (origins of gap in perf between text and vision; of exploration efficiency)
- parallels to existing work
- relationships to human performance and human reasoning
- some overstated claims

With the increase of one score from 4 to 6, a consensus for acceptance was reached. However, the other 4 reviewers did not react prior to the discussion process being frozen by the program chairs after the openreview leaks and it might be likely that further increases could have been made. The authors' answers were convincing in most (but not all aspects) of the work and included new elements, like the requested human experiments.

The remaining (somewhat major) weakness is the simplicity of the environment, on which all reviewers agreed. In spite of this problem, the AC considers that the work is of value to the community and recommends acceptance.

**Reviewer Concerns:**

Described in the meta review

**Reviewer Scores:**

Described in the meta review

---

### Decision · Program_Chairs · 2026-01-26

Accept (Poster)